

# On the size of the space spanned by a nonequilibrium state in a quantum spin lattice system

**Maurizio Fagotti**[1⋆]

**1** LPTMS, UMR 8626, CNRS, Univ. Paris-Sud, Université Paris-Saclay, 91405 Orsay, France

⋆ maurizio.fagotti@lptms.u-psud.fr

## Abstract

We consider the time evolution of a state in an isolated quantum spin lattice system with energy cumulants proportional to the number of the sites $L^d$. We compute the distribution of the eigenvalues of the time averaged state over a time window $[t_0, t_0 + t]$ in the limit of large $L$. This allows us to infer the size of a subspace that captures time evolution in $[t_0, t_0 + t]$ with an accuracy $1 - \epsilon$. We estimate the size to be $\frac{\sqrt{2\mathfrak{e}_2}}{\pi}\mathrm{erf}^{-1}(1-\epsilon)L^{d/2}t$, where $\mathfrak{e}_2$ is the energy variance per site, and $\mathrm{erf}^{-1}$ is the inverse error function.

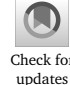

# 1 Introduction

Nonequilibrium dynamics in isolated many-body quantum systems have been being perceived as a captivating area where to look for the missing link between the physics of systems in equilibrium and the fundamental equation of quantum mechanics, the Schrödinger equation. On the one hand, the thermodynamic limit (large number of sites in spin lattice systems or large number of particles in particle systems) opens the door to critical phenomena, like spontaneous symmetry breaking and phase transitions. On the other hand, the limit of large time is characterised by a loss of information that resembles the statistical equivalence of microscopic configurations in a system in thermal equilibrium.

A weak form of relaxation can be defined by taking the time averages of the expectation values of the observables and sending the time to infinity [1]. Generally, this limit exists both when the number of degrees of freedom is finite and in the thermodynamic limit, but the non-commutativity of the limit of infinite time with the thermodynamic limit makes the definition ambiguous[1].

The time average of a nonequilibrium state has been extensively studied when the thermodynamic limit is taken after the infinite time limit, which is the setting of the quantum recurrence theorem [3]. In the opposite order of limits, the limit of infinite time usually exists in a stronger sense, without average, and indeed most of the studies have been focussed on the infinite time limit of the expectation values [4]. There is however additional information that can be extracted from the time average when the number of degrees of freedom is large. Specifically, we can use it to estimate the size of the space spanned by the state in a given time window. In this paper we show that, for a relevant class of systems characterised by extensive energy cumulants, this estimate is almost independent of the system details.

**Physical setting**

We consider a quantum spin lattice system with $L^d$ sites, $d$ being the dimension of the lattice. The system is prepared in a pure state $|\Psi_0\rangle$ that time evolves under a spin-lattice Hamiltonian $H$: $|\Psi_t\rangle = e^{-iHt}|\Psi_0\rangle$. We do not specify other details, but we assume that the energy cumulants, denoted by $L^d \mathfrak{e}_n$, are proportional to the number of the sites

$$L^d \mathfrak{e}_n = \partial_t^n \Big|_{t=0} \log \langle \Psi_0 | e^{tH} | \Psi_0 \rangle \propto L^d. \tag{1}$$

Equation (1) is satisfied in generic spin lattice systems as long as interactions and correlations decay sufficiently fast to zero with the distance[2]; in particular, if the initial state has a finite correlation length, any local (gapless or gapped) Hamiltonian satisfies (1) (see Appendix A).

Let us consider the time averaged state (see Appendix B for alternative averages):

$$\bar{\boldsymbol{\rho}}_{t_0,t} = \int_{t_0}^{t_0+t} \frac{d\tau}{t} |\Psi_\tau\rangle \langle \Psi_\tau|. \tag{2}$$

For given $L$, since the local space is finite[3], the spectrum of $H$ is discrete, and hence the infinite time limit exists; it is given by the so-called "diagonal ensemble" [1,6]

$$\lim_{t\to\infty} \bar{\boldsymbol{\rho}}_{t_0,t} = \sum_E |\langle \Psi_0 | E\rangle|^2 |E\rangle \langle E|, \tag{3}$$

where $|E\rangle$ form a basis diagonalizing $H$.

---

[1]Problems arise especially in the absence of translational invariance [2].

[2]Exceptions are known even for local Hamiltonians if the initial state has power law decaying correlations [5].

[3]If $s$ is the local spin, the dimension of the local space is $2s + 1$.

We consider here the opposite limit of finite $t$ and large $L$. We wonder how big it is the dimension $\mathfrak{D}_t$ of the space spanned by $|\Psi_0\rangle$ in the time window $[t_0, t_0 + t]$. Strictly speaking, this is given by the rank of $\bar{\boldsymbol{\rho}}_{t_0,t}$. The latter is however sensitive to infinitesimally small perturbations which do not really affect the dynamics of the physical observables. It is then more useful to approximate the state up to a given accuracy so as to reduce the dimension of the subspace, still capturing the relevant part of the dynamics. We do it at the level of the time averaged state, introducing a low-probability cutoff $\epsilon_t$, possibly dependent on the width of the time window, for the eigenvalues of $\bar{\boldsymbol{\rho}}_{t_0,t}$. This leads to the following definition

$$
\begin{aligned}
\mathfrak{D}_t^{(\epsilon_t)} &= \mathrm{tr}[\theta_H(\bar{\boldsymbol{\rho}}_{t_0,t} - \lambda_{\epsilon_t})] \\
\epsilon_t &= \mathrm{tr}[\bar{\boldsymbol{\rho}}_{t_0,t} \theta_H(\lambda_{\epsilon_t} - \bar{\boldsymbol{\rho}}_{t_0,t})],
\end{aligned}
\tag{4}
$$

where $\theta_H(x)$ is the Heaviside step function, and $\lambda_{\epsilon_t}$ is the corresponding cutoff in the eigenvalues. Note that the rank of $\bar{\boldsymbol{\rho}}_{t_0,t}$ can be obtained as a limit: $\mathfrak{D}_t = \lim_{\epsilon_t \to 0} \mathfrak{D}_t^{(\epsilon_t)}$. We will come back to the choice of $\epsilon_t$ later; we focus first on the solution of (4) for given $\epsilon_t$.

**Loschmidt echo**

Finding a solution to (4) is a hard problem, but there are crucial simplifications in the limit of large $L$. As it will be clear in the next section, these simplifications can be traced back to the behaviour of the overlap between the state at different times

$$
\langle \Psi_{t_1} | \Psi_{t_2} \rangle = \langle \Psi_0 | e^{iH(t_1 - t_2)} | \Psi_0 \rangle .
\tag{5}
$$

We remind the reader that the square of the absolute value of the overlap is also known as Loschmidt echo [7, 8] in the specific case when the backward evolution is generated by a Hamiltonian which $|\Psi_0\rangle$ is eigenstate of. The series expansion about $t_1 = t_2$ of the logarithm of the overlap can be written as follows

$$
\log \langle \Psi_{t_1} | \Psi_{t_2} \rangle = L^d \sum_{n=1}^{\infty} \frac{i^n \mathfrak{c}_n}{n!} (t_1 - t_2)^n ,
\tag{6}
$$

where, by assumption (*cf.* (1)), each order of the expansion is proportional to the number of the sites, *i.e.*, it is "extensive". In quantum spin lattice systems, extensivity is a non-perturbative feature, indeed one generally finds

$$
\exists \lim_{L \to \infty} -\frac{\log \langle \Psi_t | \Psi_0 \rangle}{L^d} \equiv f(t).
\tag{7}
$$

By definition $f(t)$ is a nonnegative function with a zero at $t = 0$. Generally it remains finite in the limit of infinite time, and it can exhibit non-analytic behavior, which has been a subject of intensive investigations since 2012 [9]. (The interested reader can find some numerical data, showing the behaviour of the Loschmidt echo in one- and two-dimensional lattices, in Refs [10, 11].) A simple but powerful property that we are going to assume and exploit is that

*exceptions apart[4], $f(t)$ has a single zero on the real line.*

In other words, the overlap is exponentially small (in the number of the sites) everywhere but in the neighbourhoods of $t = 0$. The time window where it is not exponentially small has to shrink to zero in the thermodynamic limit $L \to \infty$. In that region, the series expansion in the time (6) can also be interpreted as an asymptotic expansion in the number of the sites.

---

[4]If $f(t)$ has more than one zero, it must have infinitely many equidistant zeros, corresponding to times at which the system returns to the initial state (with potential discrepancies approaching zero in the thermodynamic limit). Generally, these are trivial situations where, for example, the energy levels are equidistant.

## 2 Entropies

We compute here the moments of the distribution of the eigenvalues of $\bar{\boldsymbol{\rho}}_{t_0,t}$, namely $\mathrm{tr}[\bar{\boldsymbol{\rho}}_{t_0,t}^{\alpha}]$ for integer $\alpha$. First of all, we note that they are independent of $t_0$, as $\bar{\boldsymbol{\rho}}_{t_0,t} = e^{-iHt_0}\bar{\boldsymbol{\rho}}_{0,t}e^{iHt_0}$ is unitarily equivalent to $\bar{\boldsymbol{\rho}}_{0,t}$ and the moments are invariant under unitary transformations. From now on we will write $\bar{\boldsymbol{\rho}}_t$ instead of $\bar{\boldsymbol{\rho}}_{0,t}$. We start with the second moment $\mathrm{tr}[\bar{\boldsymbol{\rho}}_t^2]$

$$
\mathrm{tr}[\bar{\boldsymbol{\rho}}_t^2] = \iint\limits_{[0,t]^2} \frac{\mathrm{d}\tau_2\mathrm{d}\tau_1}{t^2} |\langle\Psi_0|e^{iH(\tau_2-\tau_1)}|\Psi_0\rangle|^2 = \iint\limits_{[0,t]^2} \frac{\mathrm{d}\tau_2\mathrm{d}\tau_1}{t^2} e^{2L^d\sum_{n=1}^{\infty}\mathfrak{e}_{2n}(-1)^n\frac{(\tau_2-\tau_1)^{2n}}{(2n)!}} =
$$

$$
= \frac{2}{L^{\frac{d}{2}}}\int_0^{L^{\frac{d}{2}}}\mathrm{d}y(1-\frac{y}{L^{\frac{d}{2}}})e^{-\mathfrak{e}_2y^2t^2+2\sum_{n=2}^{\infty}\mathfrak{e}_{2n}(-1)^n\frac{(yt)^{2n}}{(2n)!L^{d(n-1)}}} , \tag{8}
$$

where we have formally replaced the function $f(t)$ with its series expansion. The contributions to the integral coming from regions where $y$ increases with $L$ are subleading[5], and, in turn, only the term proportional to $\mathfrak{e}_2$ survives the limit. We then find

$$
\mathrm{tr}[\bar{\boldsymbol{\rho}}_t^2] = \sqrt{\frac{\pi}{\mathfrak{e}_2}}t^{-1}L^{-\frac{d}{2}} + O(L^{-d}). \tag{9}
$$

We stress again that here $t$ is a finite nonzero parameter (this expression does not capture the behaviour for $t \sim L^{-\frac{d}{2}}$). From (9) we infer the asymptotic behaviour of the second Rényi entropy $S_2[\bar{\boldsymbol{\rho}}_t] = -\log\mathrm{tr}[\bar{\boldsymbol{\rho}}_t^2]$

$$
S_2[\bar{\boldsymbol{\rho}}_t] = \frac{d}{2}\log L + \frac{1}{2}\log\frac{\mathfrak{e}_2t^2}{\pi} + O(L^{-\frac{d}{2}}). \tag{10}
$$

We now compute a generic moment $\mathrm{tr}[\bar{\boldsymbol{\rho}}_t^{\alpha}]$. We have

$$
\mathrm{tr}[\bar{\boldsymbol{\rho}}_t^{\alpha}] = \int\cdots\int\limits_{[0,t]^{\alpha}} \frac{\mathrm{d}^{\alpha}\tau}{t^{\alpha}} e^{L^d\sum_{n=1}^{\infty}\mathfrak{e}_{2n}(-1)^n\frac{(\tau_{\alpha}-\tau_1)^{2n}+\sum_{j=1}^{\alpha-1}(\tau_j-\tau_{j+1})^{2n}}{(2n)!}} \times
$$

$$
\times \cos\Big(L^d\sum_{n=1}^{\infty}\mathfrak{e}_{2n+1}(-1)^n\frac{(\tau_{\alpha}-\tau_1)^{2n+1}+\sum_{j=1}^{\alpha-1}(\tau_j-\tau_{j+1})^{2n+1}}{(2n+1)!}\Big). \tag{11}
$$

The considerations made for $\alpha = 2$ hold true also for $\alpha > 2$: for any given integer $\alpha$, we can neglect the cumulants higher than the second one at the price of introducing a relative error

---

[5]This can be readily seen by splitting the integration domain into $[0,L^{\frac{d}{8}}]\cup[L^{\frac{d}{8}},L^{\frac{d}{2}}]$ and imposing that $f(t) \geq c\min(t,\delta t)^2$, for some positive finite $\delta t$ and $c$.

$O(L^{-d})$. We can then write the moments as follows[6]

$$
\text{tr}[\bar{\boldsymbol{\rho}}_t^\alpha] \sim \int \cdots \int_{[0, t\sqrt{L}]^\alpha} \frac{\mathrm{d}^\alpha \tau}{t^\alpha L^{d\frac{\alpha}{2}}} e^{-\mathfrak{e}_2 \frac{(\tau_\alpha - \tau_1)^2 + \sum_{j=1}^{\alpha-1}(\tau_j - \tau_{j+1})^2}{2}} =
$$

$$
= \frac{1}{t^\alpha L^{d\frac{\alpha}{2}}} \int_0^{tL^{\frac{d}{2}}} \mathrm{d}\tau_\alpha' \int_{-\tau_\alpha'}^{tL^{\frac{d}{2}} - \tau_\alpha'} \mathrm{d}\tau_{\alpha-1}' \int_{-\tau_{\alpha-1}' - \tau_\alpha'}^{tL^{\frac{d}{2}} - \tau_{\alpha-1}' - \tau_\alpha'} \mathrm{d}\tau_{\alpha-2}' \cdots \int_{-\sum_{j=2}^\alpha \tau_j'}^{tL^{\frac{d}{2}} - \sum_{j=2}^\alpha \tau_j'} \mathrm{d}\tau_1' e^{-\mathfrak{e}_2 \frac{(\sum_{j=1}^{\alpha-1} \tau_j')^2 + \sum_{j=1}^{\alpha-1}(\tau_j')^2}{2}} \sim
$$

$$
= \int \cdots \int_{[-\infty, \infty]^{\alpha-1}} \frac{\mathrm{d}^{\alpha-1}\tau'}{t^{\alpha-1} L^{d\frac{\alpha-1}{2}}} e^{-\mathfrak{e}_2 \frac{(\sum_{j=1}^{\alpha-1} \tau_j')^2 + \sum_{j=1}^{\alpha-1}(\tau_j')^2}{2}} = \alpha^{-\frac{1}{2}} \left(\frac{\mathfrak{e}_2}{2\pi}\right)^{\frac{1-\alpha}{2}} t^{1-\alpha} L^{d\frac{1-\alpha}{2}} . \tag{12}
$$

Thus, the asymptotic behaviour of the Rényi entropies $S_\alpha[\bar{\boldsymbol{\rho}}_t] = \frac{1}{1-\alpha} \log \text{tr}[\bar{\boldsymbol{\rho}}_t^\alpha]$ reads as

$$
S_\alpha[\bar{\boldsymbol{\rho}}_t] = \frac{d}{2} \log L + \frac{1}{2} \log \frac{\mathfrak{e}_2 t^2}{2\pi} + \frac{\log \alpha}{2(\alpha-1)} + O(L^{-\frac{d}{2}}). \tag{13}
$$

As shown in Appendix C, the leading correction $O(L^{-\frac{d}{2}})$ comes from a more careful integration over the variable that is not modulated by the gaussian. In Appendix D, our estimates for the Rényi entropies are checked agains numerics in the transverse field Ising chain.

A straightforward application of the replica trick gives the von Neumann entropy $S_{\text{vN}}[\bar{\boldsymbol{\rho}}_t] = -\text{tr}[\bar{\boldsymbol{\rho}}_t \log \bar{\boldsymbol{\rho}}_t] \overset{\text{r.t.}}{=} \lim_{\alpha \to 1} S_\alpha[\bar{\boldsymbol{\rho}}_t]$

$$
S_{\text{vN}}[\bar{\boldsymbol{\rho}}_t] \sim \frac{d}{2} \log L + \frac{1}{2} \log \frac{\mathfrak{e}_2 t^2}{2\pi} + \frac{1}{2}. \tag{14}
$$

## 3 Distribution of eigenvalues

Since the support of the distribution of eigenvalues is bounded, the moments, and, in turn, the Rényi entropies, characterise the distribution completely (Hausdorff moment problem [12]). The distribution can then be reconstructed using the approach proposed in Ref. [13]. To that aim, we define $P_{\boldsymbol{\rho}}(\lambda)$ as the (nonnormalized) distribution of the eigenvalues $\lambda_j$ of the density matrix $\boldsymbol{\rho}$: $P_{\boldsymbol{\rho}}(\lambda) = \sum_j \delta(\lambda - \lambda_j)$. It turns out that $\Phi_{\boldsymbol{\rho}}(\lambda) \equiv \lambda P_{\boldsymbol{\rho}}(\lambda)$ can be written as

$$
\Phi_{\boldsymbol{\rho}}(\lambda) = \sum_j \lambda_j \delta(\lambda - \lambda_j) = \lim_{\epsilon \to 0^+} \text{Im}\, \phi_{\boldsymbol{\rho}}(\lambda - i\epsilon), \qquad \text{with} \quad \phi_{\boldsymbol{\rho}}(z) = \frac{1}{\pi} \sum_{n=1}^\infty z^{-n} \text{tr}[\boldsymbol{\rho}^n]. \tag{15}
$$

Notwithstanding we computed only the leading order of the asymptotic expansion of the moments in the limit of large $L$, we expect the corrections to be subleading almost everywhere but close to $\lambda = 0$. Thus, we can use (12) to reconstruct the asymptotic distribution. Plugging (12) into (15) gives

$$
\phi_{\bar{\boldsymbol{\rho}}_t}(z) \sim \frac{1}{\pi} \sum_{n=1}^\infty \frac{z^{-n}}{\sqrt{n}} \left(\frac{\mathfrak{e}_2 L^d t^2}{2\pi}\right)^{\frac{1-n}{2}} = \sqrt{\frac{\mathfrak{e}_2}{2\pi^3}} L^{\frac{d}{2}} t\, \text{Li}_{1/2}\left(\sqrt{\frac{2\pi}{\mathfrak{e}_2}} \frac{1}{L^{\frac{d}{2}} tz}\right), \tag{16}
$$

---

[6]In the second line we changed integration variables into $\tau_j' = \tau_j - (1 - \delta_{j\alpha})\tau_{j+1}$. In the third line we used that, for any value of $\tau_\alpha'$ proportional to $L$, the other variables are integrated over a region surrounding 0 and increasing with $L$; the gaussian integrand makes then it possible to extend the domain of the $\alpha - 1$ variables to infinity.

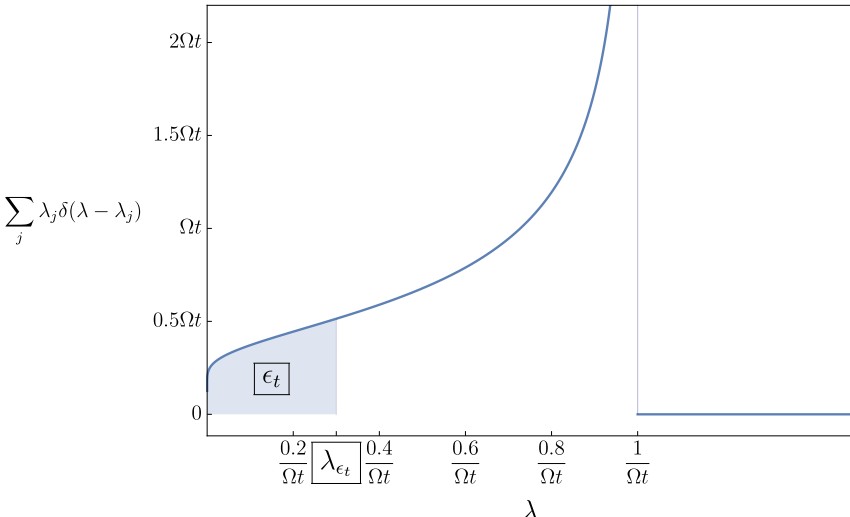

Figure 1: The asymptotic probability distribution of the eigenstates of the time averaged state in the limit of large volume; $\Omega = \sqrt{\frac{\mathfrak{e}_2}{2\pi}} L^{\frac{d}{2}}$. The shaded area has probability $\epsilon_t$.

where $\mathrm{Li}_{1/2}(x)$ is the polylogarithm of order $1/2$. The distribution of eigenvalues is then

$$P_{\bar{\rho}_t}(\lambda) = \lim_{\epsilon \to 0^+} \sqrt{\frac{\mathfrak{e}_2}{2\pi^3}} \frac{L^{\frac{d}{2}} t}{\lambda} \mathrm{Im}\,\mathrm{Li}_{\frac{1}{2}}\left( \sqrt{\frac{2\pi}{\mathfrak{e}_2}} \frac{1}{L^{\frac{d}{2}} t \lambda} + i\epsilon \right) =$$

$$= \frac{L^{\frac{d}{2}} t}{\pi \lambda} \sqrt{\frac{\mathfrak{e}_2}{\log \frac{2\pi}{\mathfrak{e}_2 L^d t^2 \lambda^2}}} \theta_H \left( \sqrt{\frac{2\pi}{\mathfrak{e}_2}} \frac{1}{L^{\frac{d}{2}} t} - \lambda \right), \quad (17)$$

where we used that $\lim_{\epsilon \to 0^+} \mathrm{Li}_{\nu}(x + i\epsilon) = \pi (\log x)^{\nu-1}/\Gamma(\nu) \theta_H(x-1)$, with $\Gamma(x)$ the gamma function. Note that, by definition, $P_{\bar{\rho}_t}(\lambda)$ is normalised to the dimension of the Hilbert space; the asymptotic result (17), on the other hand, despite capturing all the moments $\int_0^1 d\lambda P_{\bar{\rho}_t}(\lambda) \lambda^n$ with integer $n > 0$, has a divergent integral. This is because the behavior for $\lambda \sim o(L^{-\frac{d}{2}})$ goes beyond the leading order of the asymptotic expansion.

The distribution $\Phi_{\bar{\rho}_t}(\lambda)$, shown in figure 1, is instead correctly normalised

$$d\lambda \Phi_{\bar{\rho}_t}(\lambda) = d\left[ \sqrt{\frac{\mathfrak{e}_2}{2\pi}} L^{\frac{d}{2}} t \lambda \right] \Pi\left( \sqrt{\frac{\mathfrak{e}_2}{2\pi}} L^{\frac{d}{2}} t \lambda \right), \qquad \text{with} \quad \Pi(x) \sim \frac{\theta_H(1-x)}{\sqrt{-\pi \log x}}. \quad (18)$$

Again, we expect the corrections to this asymptotic result to mainly affect the behaviour of $\Pi(x)$ close to $x = 0$.

In view of the universality of (18), we come to the conclusion that the large $L$ behaviour of the distribution of the eigenvalues of a time averaged state is a fundamental feature of quantum spin lattice systems with extensive energy cumulants.

# 4 Effective rank of the time averaged state

Having computed the asymptotic behaviour of $\Phi_{\bar{\rho}_t}(\lambda)$ and of $P_{\bar{\rho}_t}(\lambda)$ in the limit of large number of sites, we have access to the asymptotic solution of (4), provided that $\lambda_{\epsilon_t} \sim O(L^{-\frac{d}{2}})$.

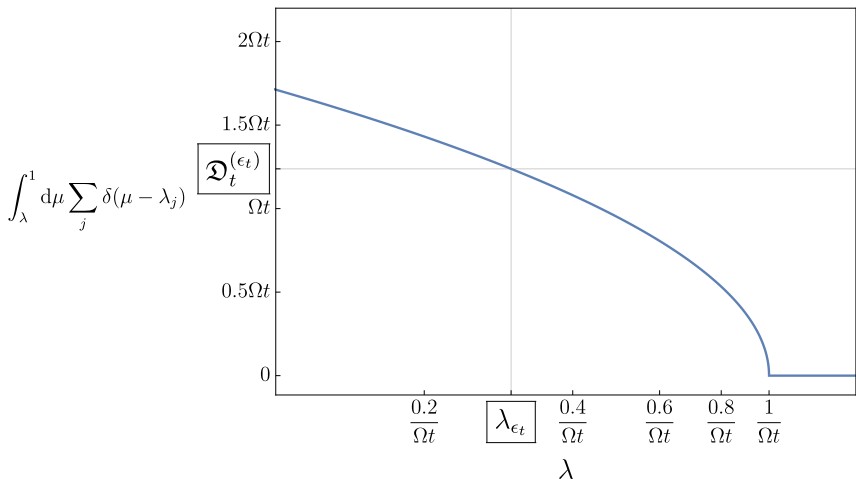

Figure 2: The asymptotic number of eigenvalues larger than $\lambda$ for a lattice with a large number of sites; $\Omega = \sqrt{\frac{\mathfrak{c}_2}{2\pi}} L^{\frac{d}{2}}$. The point $(\lambda_{\epsilon_t}, \mathfrak{D}_t^{(\epsilon_t)})$ identifies a subspace that captures the averaged state with error $\epsilon_t$ (*cf* figure 1). (Note that the x-axis is shown in logarithmic scale.)

From (18) it follows

$$\epsilon_t \sim \int_0^{x_{\epsilon_t}} dx \frac{1}{\sqrt{-\pi \log x}} = 1 - \mathrm{erf}\left(\sqrt{-\log x_{\epsilon_t}}\right)$$

$$\mathfrak{D}_t^{(\epsilon_t)} \sim \sqrt{\frac{\mathfrak{c}_2}{2\pi}} L^{\frac{d}{2}} t \int_{x_{\epsilon_t}}^1 \frac{dx}{x} \frac{1}{\sqrt{-\pi \log x}} = \frac{\sqrt{2\mathfrak{c}_2}}{\pi} \sqrt{-\log x_{\epsilon_t}} L^{\frac{d}{2}} t = \frac{\sqrt{2\mathfrak{c}_2}}{\pi} \mathrm{erf}^{-1}(1 - \epsilon_t) L^{\frac{d}{2}} t \,,$$

(19)

which are expected to be valid as long as $\epsilon_t$ does not approach zero in the thermodynamic limit. Figures 1 and 2 provide a graphic representation of $\epsilon_t$ and $\mathfrak{D}_t^{(\epsilon_t)}$. Assuming $\epsilon_t$ small, we can expand the inverse error function as follows

$$\mathrm{erf}^{-1}(1 - \epsilon_t) \overset{\epsilon_t \ll 1}{\sim} \sqrt{\frac{\log \frac{2}{\pi \epsilon_t^2} - \log \log \frac{2}{\pi \epsilon_t^2}}{2}} \,,$$

(20)

so we find

$$\mathfrak{D}_t^{(\epsilon_t)} \approx \frac{\sqrt{\mathfrak{c}_2}}{\pi} \sqrt{-\log\left(-\frac{\pi \epsilon_t^2}{2} \log \frac{\pi \epsilon_t^2}{2}\right)} L^{\frac{d}{2}} t \,.$$

(21)

The next step is to establish a connection between the cutoff $\epsilon_t$ and the error on the state. To that aim, let us introduce a tiny time scale $\delta t$ such that $t/\delta t$ is integer. The time average in $[0, t]$ can be seen as the mean of the sample consisting of the time averaged states over the time windows $[(n-1)\delta t, n\delta t]$, with $n \in [1, t/\delta t]$. For asymptotically large $L$, one could naively expect the errors produced by truncating the spectrum of $\bar{\rho}_t$ to be randomly distributed over the sample. Under this assumption, since the variance of independent random variables is additive, the error $\epsilon_t$ of the mean of the sample would scale as $\epsilon_t \sim \epsilon_{\delta t} \sqrt{\delta t / t}$, where $\epsilon_{\delta t}$ is

the truncation error in each time slice. This gives

$$\mathfrak{D}^{(\epsilon_{\delta t})}_{t|\delta t} \sim \frac{\sqrt{2\mathfrak{e}_2}}{\pi} \mathrm{erf}^{-1}\left(1 - \epsilon_{\delta t}\sqrt{\frac{\delta t}{t}}\right) L^{\frac{d}{2}} t \approx \frac{\sqrt{\mathfrak{e}_2}}{\pi}\sqrt{\log\left(\frac{2t}{\pi\epsilon_{\delta t}^2 \delta t \log\frac{2t}{\pi\epsilon_{\delta t}^2 \delta t}}\right)} L^{\frac{d}{2}} t. \tag{22}$$

We are interested in observables that have a well-defined expectation value in the thermodynamic limit (for example, local observables), for which the variation of their expectation value in a tiny time window approaches zero as the width of the interval shrinks to zero. Thus, if $\delta t$ is small enough, $\epsilon_{\delta t}$ can be identified with the effective error on the time evolving state, and $\mathfrak{D}^{(\epsilon_{\delta t})}_{t|\delta t}$ is the effective dimension we are looking for. This observable-dependent step gives a meaning to "relevance": our truncation would be inadequate if we would consider observables that have different expectation values in $|\Psi_t\rangle$ and $|\Psi_{t+\tau_L}\rangle$, with $\lim_{L\to\infty} \tau_L = 0$. Note also that we are not allowed to choose a cutoff $\delta t$ approaching zero in the thermodynamic limit because that would require the knowledge of the next orders of the asymptotic expansion, which is trickier (see Appendix C). Appendix D includes some numerical checks of (22) in generic spin chains.

Under the assumption of independence, the projection on a space with size (22) is arguably the best approximation with error $\epsilon_{\delta t}$ if we have only access to the time averaged state; but this is far from being optimal. It is more convenient to merge the reduced spaces in the time slices $[(n-1)\delta t, n\delta t]$, each of which is the span of $\mathfrak{D}^{(\epsilon_{\delta t})}_{\delta t}$ states. The size of the resulting space is bounded from above by the total number of elements, which is $\sum_{n=1}^{t/\delta t} \mathfrak{D}^{(\epsilon_{\delta t})}_{\delta t} = \mathfrak{D}^{(\epsilon_{\delta t})}_{t}$. In the limit $\delta t \to 0$, we reinterpret $\epsilon_{\delta t} \to \epsilon$ as the truncation error on the state[7]; we finally find the upper bound $\mathfrak{D}^{(\epsilon)}_{t}$, which is tighter than (22). This is not the end of the story. Since the error on the time average can not be larger than the error on the state and $\mathfrak{D}^{(\epsilon)}_{t}$ is also the size of the space capturing the time average with error $\epsilon$, we conclude that the upper bound is asymptotically saturated (which means, in turn, that the assumption of independence that we made before is not satisfied). We can now state our main result:

*The size of the space that is approximately spanned by a nonequilibrium state in the time window $[t_0, t_0 + t]$ with error $\epsilon$ on the state is asymptotically given by*

$$\mathfrak{D}^{(\epsilon)}_{t} \sim \frac{\sqrt{2\mathfrak{e}_2}}{\pi} \mathrm{erf}^{-1}(1-\epsilon) L^{\frac{d}{2}} t. \tag{23}$$

## 4.1 Numerical simulations of nonequilibrium dynamics

The result (23) is rather suggestive if reconsidered in the context of simulations of out-of-equilibrium many body quantum systems. First of all, the relevant space where the dynamics take place is proportional to the square root of the logarithm of the Hilbert space. In addition, within the assumptions of our calculation, a Lieb-Robinson velocity $v_{\mathrm{LR}}$ generally exists [14,15], which bounds the speed at which information propagates throughout the lattice. Consequently, the dynamics of a compact subsystem of size $\ell^d$ in the time window $[0, t]$ display exponentially small finite-size effects, provided that $L \gtrsim \ell + 2v_{\mathrm{LR}}t$: we can replace $L$ by $\ell + 2v_{\mathrm{LR}}t + R$ in (23), making an error that is exponentially small in $R$. In conclusion, in the limit of large time, the size of the relevant subspace does not grow faster than $\sim \sqrt{\mathfrak{e}_2} v_{\mathrm{LR}}^{\frac{d}{2}} t^{\frac{d}{2}+1}$. This provides a physical reference value for the time step $\delta t$ to choose in numerical simulations: if we identify the "frame rate" of the time evolving state simulated with the rate at

---

[7] As in the previous discussion, we are restricting our attention to the system properties compatible with $|\Psi_t\rangle\langle\Psi_t| \overset{L\to\infty}{\sim} \lim_{\delta t\to 0}\lim_{L\to\infty} \bar{\rho}_{t,\delta t}$, which would have been an identity if the order of limits were reversed.

which the relevant subspace capturing the dynamics of local observables increases, we obtain $\delta t \sim (v_{\text{LR}} t)^{-\frac{d}{2}}/\sqrt{\mathfrak{c}_2}$. This formula could be used, for example, to reset the time step of a simulation when a different system is considered.

Finally, we note that an algorithm reducing the dynamics onto the relevant subspace would allow for investigations in much wider time windows than those accessible nowadays through state-of-the-art techniques (which, in spin chains, could be time-dependent density matrix renormalisation group (tDRMG) [16] and infinite time-evolving block decimation (iTEBD) [17] algorithms).

### 4.2 Quantum speed limit

Our findings are complementary to the studies on the minimum time $\tau$ required for arriving to an orthogonal state - "the quantum speed limit"; we mention here just the classical result by Mandelstam and Tamm [18] $\tau \geq \pi L^{-d/2}/2\sqrt{\mathfrak{c}_2}$[8]. Notwithstanding the similarity between this limit and our estimate, their meaning is quite different. In our approach a new orthogonal state starts counting in $\mathfrak{D}_t^{(\epsilon)}$ after developing a significant overlap, not necessarily equal to 1, with the time evolving state. In addition, the analogue of the quantum speed limit in the quantum systems considered is the time needed to have a so-called "dynamical phase transition" [9]. Generally, the latter time remains nonzero even in the thermodynamic limit, and, in a hypothetical case where it does not, the hypotheses behind the asymptotic expansion that we carried out would not be fulfilled.

## 5 Conclusion

We studied the nonequilibrium time evolution of a state under a Hamiltonian of a general quantum spin lattice system with energy cumulants proportional to the number of the sites. This is a mild condition that is always fulfilled whenever the Hamiltonian is (quasi)local and the state has finite correlation lengths - see Appendix A. We have computed the leading order of the asymptotic expansion of the distribution of the eigenvalues of the time averaged state over a fixed time window in the limit of a large number of sites. We used the asymptotic distribution to determine the size of the space visited by the state, and we have found that it is proportional to the square root of the logarithm of the Hilbert space. It would be interesting to generalise our results to the time evolution of critical states with super-extensive energy cumulants, like the ones considered in Ref. [5]. Finally, our estimate does not distinguish chaotic systems from integrable ones (see also Appendix D), which are expected to time evolve with a lower complexity [20] (see also [21] and references therein for more recent investigations); this points to the existence of differences in the properties of the eigenvectors of the time averaged state, which in generic systems are apparently badly approximated by matrix product states even when the initial state has fast decaying correlations and the Hamiltonian is local.

## Acknowledgements

These notes have been prepared for the course "Quench dynamics and relaxation in isolated integrable quantum spin chains", held in IPhT Saclay from 25/01/2019 to 22/02/2019. I thank Grégoire Misguich for useful discussions.

---

[8]For the class of systems that we have considered, the more recent finding by Margolus and Levitin [19] is not as tight as the Mandelstam and Tamm bound.

**Funding information** This work was supported by a grant LabEx PALM (ANR-10-LABX-0039-PALM) and by the European Research Council under the Starting Grant No. 805252 LoCoMacro.

## A  Are the energy cumulants extensive?

In this appendix we discuss the hypothesis that the energy cumulants are extensive. We use the following definitions:

- $O$ is *localised* if it acts like the identity everywhere but on a compact subsystem. The latter is called "support" of $O$.

- $O$ is *quasilocalised* if it can be approximated by a localised operator and the error made decays exponentially with the extent of the support of the localised operator.

- $A$ is *(quasi)local* if $i[A, O]$ is (quasi)localized for every localized operator $O$.

For the sake of simplicity we focus on spin chains described by quasilocal Hamiltonians $H$, but we do not expect significant differences in higher dimensional lattice systems. Let $|\Psi_0\rangle$ be the initial state. We define the "imaginary time evolving state" as

$$|\Psi_\beta\rangle = \frac{e^{\frac{\beta}{2}H}}{\sqrt{\langle \Psi_0|e^{\beta H}|\Psi_0\rangle}} |\Psi_0\rangle \,. \tag{24}$$

One can readily show that the energy cumulants can be obtained as follows

$$L\mathfrak{e}_n = \sum_\ell \mathfrak{e}_n(\ell) \qquad \text{with} \quad \mathfrak{e}_n(\ell) = \partial_\beta^{n-1}\Big|_{\beta=0} \langle \Psi_\beta|\boldsymbol{h}_\ell|\Psi_\beta\rangle \,, \tag{25}$$

where $\boldsymbol{h}_\ell$ is the energy density about a given site $\ell$, defined in such a way that $H = \sum_\ell \boldsymbol{h}_\ell$. The quantities $\mathfrak{e}_n(\ell)$ will be referred to energy cumulant densities. If we can interpret $|\Psi_\beta\rangle$ as the ground state of a quasilocal (Hermitian) Hamiltonian $H_\beta$, we immediately see that a cumulant per unit length can diverge only if there is a quantum phase transition at $\beta = 0$; in that case, $|\Psi_0\rangle$ is the ground state of a critical system, and it is expected to have power-law decaying correlations.

In order to be more quantitative, it is convenient to represent the energy cumulant densities as connected correlations in the state $|\Psi_0\rangle$

$$\mathfrak{e}_n(\ell) = \langle H^{(n)}\boldsymbol{h}_\ell\rangle - \langle H^{(n)}\rangle \langle \boldsymbol{h}_\ell\rangle \,, \tag{26}$$

where $H^{(n)}$ are given by

$$H^{(n)} = \partial_\beta^n\Big|_{\beta=0} \frac{e^{\beta H} - 1}{\langle e^{\beta H}\rangle} \,. \tag{27}$$

We also report a recursive definition

$$H^{(n)} = H^n - \sum_{j=1}^{n-1} \binom{n}{j} \langle H^{n-j}\rangle H^{(j)} \equiv H^n - \sum_{j=1}^{n-1} \binom{n}{j} \langle H^{(n-j)}\rangle H^j \,. \tag{28}$$

For example, we have

$$\begin{aligned}
H^{(1)} &= H \\
H^{(2)} &= H^2 - 2\langle H\rangle H \\
H^{(3)} &= H^3 - 3\langle H\rangle H^2 - 3(\langle H^2\rangle - 2\langle H\rangle^2)H \\
H^{(4)} &= H^4 - 4\langle H\rangle H^3 - 6(\langle H^2\rangle - 2\langle H\rangle^2)H^2 - 4(\langle H^3\rangle - 6\langle H^2\rangle\langle H\rangle + 6\langle H\rangle^3)H \,.
\end{aligned} \tag{29}$$

**Lemma 1.** *If $|\Psi_0\rangle$ has exponentially decaying correlations, the connected correlation between $H^{(n)}$ and a generic localised operator $O$ is close to the one between the quasilocalised operator $H_S^{(n)}$ and $O$*

$$\langle H^{(n)}O\rangle - \langle H^{(n)}\rangle\langle O\rangle \sim \langle H_S^{(n)}O\rangle - \langle H_S^{(n)}\rangle\langle O\rangle\,, \tag{30}$$

*where*

$$H_S^{(n)} = \partial_\beta^n\Big|_{\beta=0}\frac{e^{\beta H_S}-1}{\langle\Psi_0|e^{\beta H_S}|\Psi_0\rangle}\,, \tag{31}$$

*and $H_S$ is the truncated Hamiltonian*

$$H_S = \sum_{\ell\in S} h_\ell\,. \tag{32}$$

*Here $S$ is a subsystem that contains the support of $O$; if $S$ is enlarged in such a way that the support of $O$ remains in the bulk of $S$, the error made decreases exponentially with the extent $|S|$ of $S$.*

By this lemma, the cumulants (26) are expressed as a sum of connected correlations between quasilocalised operators; since such correlations can not diverge, we can state the following

**Proposition.** *The energy cumulants of a quasilocal Hamiltonian in a state with a finite correlation length are extensive. This result holds true even in the absence of translational invariance and independently of whether the Hamiltonian is critical or not.*

On the other hand, in the presence of power-law decaying correlations, we expect some energy cumulants to scale differently with the system size. Ref. [5] provided as example of this anomalous behaviour in the second energy cumulant considering the quantum Ising model.

We refer the reader to Refs [22,23] for closely related results; we present here a constructive proof of Lemma 1.

Without loss of generality, we can assume $\langle O\rangle \equiv \langle\Psi_0|O|\Psi_0\rangle=0$, so the connected correlation can be identified with the correlation. Let $A = \sum_\ell a_\ell$ be a quasilocal operator. We define $A_\bullet$ as a truncation of $A$ with support including the support of $O$ ($A_\bullet = \sum_{\ell\in S} a_\ell$ for some set $S$ containing the support of $O$) and with $A_\circ$ the rest ($A_\circ = \sum_{\ell\notin S} a_\ell$). We note that generally $A_{\bullet\circ}$ does not commute with $A_\circ$, whereas the commutator between $A_{\bullet\bullet}$ and $A_\circ$ can be made arbitrarily small by enlarging the subsystems. A finite correlation length implies that the connected correlation between $A$ and a localised operator $O$ is exponentially close to the one between $A_\bullet$ and $O$

$$\langle AO\rangle = \langle A_\bullet O\rangle + O(e^{-|S|/\xi})\,. \tag{33}$$

From now on, every time that a finite set of compact subsystems can be chosen in such a way that an equation is valid up to exponentially small corrections in the extent of a subsystem, we will use the symbol $\sim$. In particular, we have obtained $\langle AO\rangle \sim \langle A_\bullet O\rangle$, and hence

$$\langle H^{(1)}O\rangle = \langle HO\rangle \sim \langle H_\bullet O\rangle = \langle H_\bullet^{(1)}O\rangle\,, \tag{34}$$

which is the first local identity stated in Lemma 1 ($n = 1$).

More generally, in order to exploit the finiteness of the correlation length in a consistent way, we introduce a canonical decomposition where the expressions are written in such a way that either all the operators appearing in the expectation values have a single $\bullet$, or they do not contain the support of $O$ at all (so they have a final $\circ$). For example we have

$$\begin{aligned}\langle AB\rangle = {}&\langle A_\bullet B_\bullet\rangle + \langle A_\circ B_\circ\rangle + \langle A_\bullet B_\circ\rangle + \langle A_\circ B_\bullet\rangle \sim \langle A_\bullet B_\bullet\rangle + \langle A_\circ B_\circ\rangle + \\&+ \langle A_{\bullet\bullet}\rangle\langle B_\circ\rangle + \langle A_{\bullet\circ}B_\circ\rangle + \langle A_\circ\rangle\langle B_{\bullet\bullet}\rangle + \langle A_\circ B_{\bullet\circ}\rangle = \langle A_\bullet B_\bullet\rangle + \langle A_\circ B_\circ\rangle + \\&+ \langle A_\bullet\rangle\langle B_\circ\rangle - \langle A_{\bullet\circ}\rangle\langle B_\circ\rangle + \langle A_{\bullet\circ}B_\circ\rangle + \langle A_\circ\rangle\langle B_\bullet\rangle - \langle A_\circ\rangle\langle B_{\bullet\circ}\rangle + \langle A_\circ B_{\bullet\circ}\rangle\,. \tag{35}\end{aligned}$$

For future convenience, we also introduce the notation $\{A_1, \ldots, A_n\}$ to indicate the symmetrised product of the operators $A_j$, *e.g.*

$$\{A_1, A_2, A_3\} = A_1 A_2 A_3 + A_1 A_3 A_2 + A_2 A_1 A_3 + A_2 A_3 A_1 + A_3 A_1 A_2 + A_3 A_2 A_1 \,. \tag{36}$$

If the same operator appears more than once in the symmetrised product, we write its multiplicity below a horizontal brace, *e.g.*,

$$\{A_1, \underbrace{A_2}_{2}\} = \{A_1, A_2, A_2\} \,. \tag{37}$$

Before proving Lemma 1, we provide evidence of its validity by working out the cases $n = 2, 3, 4$. This will be useful to understand the subsequent proof. The impatient reader can however skip the next sections and continue reading from Section (A.1).

**Check of $H^{(2)}$**

The canonical decomposition of $\langle H^2 O \rangle$ reads

$$\langle H^2 O \rangle \sim \langle H_\bullet^2 + \{H_\bullet, H_\circ\} O \rangle \sim \langle H_\bullet^2 O \rangle + \langle \{H_{\bullet\bullet} + H_{\bullet\circ}, H_\circ\} O \rangle \sim$$
$$\sim \langle H_\bullet^2 O \rangle + 2 \langle H_\circ \rangle \langle H_{\bullet\bullet} O \rangle \sim \langle H_\bullet^2 O \rangle + 2 \langle H_\circ \rangle \langle H_\bullet O \rangle \,. \tag{38}$$

Since $H_\circ = H - H_\bullet$, we readily obtain the second local identity

$$\langle H^{(2)} O \rangle \sim \langle H_\bullet^{(2)} O \rangle \,. \tag{39}$$

Incidentally, by inverting (38) after having replaced $H$ by $H_\bullet$, we find

$$\langle H_{\bullet\bullet}^2 O \rangle \sim \langle H_\bullet^2 O \rangle - 2 \langle H_{\bullet\circ} \rangle \langle H_\bullet O \rangle \,, \tag{40}$$

which will be useful in the following. Analogously, we have

$$\langle H^2 \rangle \sim \langle H_\bullet^2 \rangle + \langle H_\circ^2 \rangle + 2 \langle H_\bullet \rangle \langle H_\circ \rangle + \langle \{H_{\bullet\circ}, H_\circ\} \rangle - 2 \langle H_{\bullet\circ} \rangle \langle H_\circ \rangle \,. \tag{41}$$

**Check of $H^{(3)}$**

The canonical decomposition of $\langle H^3 O \rangle$ reads

$$\langle H^3 O \rangle \sim \langle (H_\bullet^3 + \frac{1}{2}\{H_\bullet, \underbrace{H_\circ}_{2}\} + \frac{1}{2}\{\underbrace{H_\bullet}_{2}, H_\circ\}) O \rangle \sim$$
$$\sim \langle H_\bullet^3 O \rangle + 3 \langle H_\circ^2 \rangle \langle H_{\bullet\bullet} O \rangle + 3 \langle H_\circ \rangle \langle H_{\bullet\bullet}^2 O \rangle + 3 \langle \{H_{\bullet\circ}, H_\circ\} \rangle \langle H_{\bullet\bullet\bullet} O \rangle \sim$$
$$\sim \langle H_\bullet^3 O \rangle + 3 (\langle H_\circ^2 \rangle + \langle \{H_{\bullet\circ}, H_\circ\} \rangle) \langle H_\bullet O \rangle + 3 \langle H_\circ \rangle \langle (H_\bullet - H_{\bullet\circ})^2 O \rangle \sim$$
$$\sim \langle H_\bullet^3 O \rangle + 3 \langle H_\circ \rangle \langle H_\bullet^2 O \rangle + 3 (\langle H_\circ^2 \rangle + \langle \{H_{\bullet\circ}, H_\circ\} \rangle - 2 \langle H_\circ \rangle \langle H_{\bullet\circ} \rangle) \langle H_\bullet O \rangle \,, \tag{42}$$

which, by virtue of (41), can also be written as

$$\langle H^3 O \rangle \sim \langle H_\bullet^3 O \rangle + 3 \langle H_\circ \rangle \langle H_\bullet^2 O \rangle + 3 (\langle H^2 \rangle - \langle H_\bullet^2 \rangle - 2 \langle H_\bullet \rangle \langle H_\circ \rangle) \langle H_\bullet O \rangle \,. \tag{43}$$

Incidentally, (42) also implies

$$\langle H_{\bullet\bullet}^3 O \rangle \sim \langle H_\bullet^3 O \rangle - 3 \langle H_{\bullet\circ} \rangle \langle H_\bullet^2 O \rangle -$$
$$- 3 (\langle H_{\bullet\circ}^2 \rangle + \langle \{H_{\bullet\bullet\circ}, H_{\bullet\circ}\} \rangle - 2 \langle H_{\bullet\circ} \rangle \langle H_{\bullet\bullet\circ} \rangle - 2 \langle H_{\bullet\circ} \rangle^2) \langle H_\bullet O \rangle \,. \tag{44}$$

The third local identity is readily checked

$$\langle H^{(3)}O\rangle = \langle H^3O\rangle - 3\langle H\rangle\langle H^2O\rangle - 3(\langle H^2\rangle - 2\langle H\rangle^2)\langle HO\rangle \sim$$
$$\sim \langle H_\bullet^3 O\rangle + 3\langle H_\circ\rangle\langle H_\bullet^2 O\rangle + 3(\langle H_\bullet^2\rangle + \langle\{H_{\bullet\circ},H_\circ\}\rangle - 2\langle H_\circ\rangle\langle H_{\bullet\circ}\rangle)\langle H_\bullet O\rangle -$$
$$- 3\langle H\rangle(\langle H_\bullet^2 O\rangle + 2\langle H_\circ\rangle\langle H_\bullet O\rangle) - 3(\langle H^2\rangle - 2\langle H\rangle^2)\langle H_\bullet O\rangle \sim$$
$$\sim \langle H_\bullet^3 O\rangle - 3\langle H_\bullet\rangle\langle H_\bullet^2 O\rangle - 3(\langle H_\bullet^2\rangle - 2\langle H_\bullet\rangle^2)\langle H_\bullet O\rangle = \langle H_\bullet^{(3)}O\rangle . \quad (45)$$

Analogously, we obtain

$$\langle H^3\rangle \sim \langle H_\bullet^3\rangle + \langle H_\circ^3\rangle + 3\langle H_\circ\rangle\Big(\langle H_\bullet^2\rangle - \langle H_{\bullet\circ}^2\rangle - 2(\langle H_\bullet\rangle - \langle H_{\bullet\circ}\rangle - \langle H_{\bullet\bullet\circ}\rangle)\langle H_{\bullet\circ}\rangle -$$
$$- \langle\{H_{\bullet\bullet\circ},H_{\bullet\circ}\}\rangle\Big) + 3\langle\{H_{\bullet\circ},H_\circ\}\rangle(\langle H_\bullet\rangle - \langle H_{\bullet\circ}\rangle - \langle H_{\bullet\bullet\circ}\rangle) + \langle\{H_{\bullet\bullet\circ},H_{\bullet\circ},H_\circ\}\rangle +$$
$$+ \frac{1}{2}\langle\{\underbrace{H_{\bullet\circ}}_{2},H_\circ\}\rangle + 3\langle H_\circ^2\rangle(\langle H_\bullet\rangle - \langle H_{\bullet\circ}\rangle) + \frac{1}{2}\langle\{H_{\bullet\circ},\underbrace{H_\circ}_{2}\}\rangle . \quad (46)$$

**Check of $H^{(4)}$**

The verification of the fourth local identity is more cumbersome, but it could be useful to dispel doubts upon the validity of Lemma 1, as the first three cases could lack some potentially dangerous structure. The first step towards the canonical decomposition of $\langle H^4 O\rangle$ reads

$$\langle H^4 O\rangle \sim \langle H_\bullet^4 O\rangle + \frac{1}{6}\langle\{H_\bullet,\underbrace{H_\circ}_{3}\}O\rangle + \frac{1}{4}\langle\{\underbrace{H_\bullet}_{2},\underbrace{H_\circ}_{2}\}O\rangle + \frac{1}{6}\langle\{\underbrace{H_\bullet}_{3},H_\circ\}O\rangle . \quad (47)$$

Let us work out term by term:

$$\frac{1}{6}\langle\{H_\bullet,\underbrace{H_\circ}_{3}\}O\rangle \sim 4\langle H_\circ^3\rangle\langle H_{\bullet\bullet}O\rangle \sim 4\langle H_\circ^3\rangle\langle H_\bullet O\rangle \quad (48)$$

$$\frac{1}{4}\langle\{\underbrace{H_\bullet}_{2},\underbrace{H_\circ}_{2}\}O\rangle \sim 6\langle H_\circ^2\rangle\langle H_{\bullet\bullet}^2 O\rangle + \frac{1}{2}\langle\{H_{\bullet\bullet},H_{\bullet\circ},\underbrace{H_\circ}_{2}\}O\rangle \sim$$
$$\sim 6\langle H_\circ^2\rangle(\langle H_\bullet^2 O\rangle - 2\langle H_{\bullet\circ}\rangle\langle H_\bullet O\rangle) + 2\langle\{H_{\bullet\circ},\underbrace{H_\circ}_{2}\}\rangle\langle H_{\bullet\bullet\bullet}O\rangle \sim$$
$$\sim 6\langle H_\circ^2\rangle\langle H_\bullet^2 O\rangle + 2(\langle\{H_{\bullet\circ},\underbrace{H_\circ}_{2}\}\rangle - 6\langle H_\circ^2\rangle\langle H_{\bullet\circ}\rangle)\langle H_\bullet O\rangle \quad (49)$$

$$\frac{1}{6}\langle\{\underbrace{H_\bullet}_{3},H_\circ\}O\rangle \sim 4\langle H_\circ\rangle\langle H_{\bullet\bullet}^3 O\rangle + \frac{1}{2}\langle\{\underbrace{H_{\bullet\bullet}}_{2},H_{\bullet\circ},H_\circ\}O\rangle + \frac{1}{2}\langle\{H_{\bullet\bullet},\underbrace{H_{\bullet\circ}}_{2},H_\circ\}O\rangle \sim$$
$$\sim 4\langle H_\circ\rangle\Big(\langle H_\bullet^3 O\rangle - 3\langle H_{\bullet\circ}\rangle\langle H_\bullet^2 O\rangle - 3(\langle H_{\bullet\circ}^2\rangle + \langle\{H_{\bullet\bullet\circ},H_{\bullet\circ}\}\rangle -$$
$$- 2\langle H_{\bullet\circ}\rangle\langle H_{\bullet\bullet\circ}\rangle - 2\langle H_{\bullet\circ}\rangle\langle H_{\bullet\circ}\rangle)\langle H_\bullet O\rangle\Big) + 6\langle\{H_{\bullet\circ},H_\circ\}\rangle\langle H_{\bullet\bullet\bullet}^2 O\rangle +$$
$$+ 4\langle\{H_{\bullet\bullet\circ},H_{\bullet\circ},H_\circ\}\rangle\langle H_{\bullet\bullet\bullet\bullet}O\rangle + 2\langle\{\underbrace{H_{\bullet\circ}}_{2},H_\circ\}\rangle\langle H_{\bullet\bullet\bullet}O\rangle \sim$$
$$\sim 4\langle H_\circ\rangle\Big(\langle H_\bullet^3 O\rangle - 3\langle H_{\bullet\circ}\rangle\langle H_\bullet^2 O\rangle - 3(\langle H_{\bullet\circ}^2\rangle + \langle\{H_{\bullet\bullet\circ},H_{\bullet\circ}\}\rangle -$$
$$- 2\langle H_{\bullet\circ}\rangle\langle H_{\bullet\bullet\circ}\rangle - 2\langle H_{\bullet\circ}\rangle\langle H_{\bullet\circ}\rangle)\langle H_\bullet O\rangle\Big) + 6\langle\{H_{\bullet\circ},H_\circ\}\rangle\Big(\langle H_\bullet^2 O\rangle -$$
$$- 2\langle H_{\bullet\circ}\rangle\langle H_\bullet O\rangle - 2\langle H_{\bullet\bullet\circ}\rangle\langle H_\bullet O\rangle\Big) + 4\langle\{H_{\bullet\bullet\circ},H_{\bullet\circ},H_\circ\}\rangle\langle H_\bullet O\rangle +$$
$$+ 2\langle\{\underbrace{H_{\bullet\circ}}_{2},H_\circ\}\rangle\langle H_\bullet O\rangle = 4\langle H_\circ\rangle\langle H_\bullet^3 O\rangle + 6\Big(\langle\{H_{\bullet\circ},H_\circ\}\rangle -$$

$$-2\langle H_\circ\rangle\langle H_{\bullet\circ}\rangle\Big)\langle H_\bullet^2 O\rangle + 2\Big(2\langle\{H_{\bullet\bullet\circ},H_{\bullet\circ},H_\circ\}\rangle + \langle\{\underbrace{H_{\bullet\circ}}_{2},H_\circ\}\rangle-$$

$$-6\langle H_\circ\rangle(\langle H_{\bullet\circ}^2\rangle + \langle\{H_{\bullet\bullet\circ},H_{\bullet\circ}\}\rangle - 2\langle H_{\bullet\circ}\rangle\langle H_{\bullet\bullet\circ}\rangle - 2\langle H_{\bullet\circ}\rangle^2)-$$

$$-6\langle\{H_{\bullet\circ},H_\circ\}\rangle(\langle H_{\bullet\circ}\rangle + \langle H_{\bullet\bullet\circ}\rangle)\Big)\langle H_\bullet O\rangle. \tag{50}$$

Putting all together we find

$$\langle H^4 O\rangle \sim \langle H_\bullet^4 O\rangle + 4\langle H_\circ\rangle\langle H_\bullet^3 O\rangle + 6\Big(\langle H_\circ^2\rangle + \langle\{H_{\bullet\circ},H_\circ\}\rangle - 2\langle H_\circ\rangle\langle H_{\bullet\circ}\rangle\Big)\langle H_\bullet^2 O\rangle +$$

$$+2\Big(2\langle H_\circ^3\rangle + \langle\{H_{\bullet\circ},\underbrace{H_\circ}_{2}\}\rangle - 6\langle H_\circ^2\rangle\langle H_{\bullet\circ}\rangle + 2\langle\{H_{\bullet\bullet\circ},H_{\bullet\circ},H_\circ\}\rangle + \langle\{\underbrace{H_{\bullet\circ}}_{2},H_\circ\}\rangle-$$

$$-6\langle H_\circ\rangle(\langle H_{\bullet\circ}^2\rangle + \langle\{H_{\bullet\bullet\circ},H_{\bullet\circ}\}\rangle - 2\langle H_{\bullet\circ}\rangle\langle H_{\bullet\bullet\circ}\rangle - 2\langle H_{\bullet\circ}\rangle^2)-$$

$$-6\langle\{H_{\bullet\circ},H_\circ\}\rangle(\langle H_{\bullet\circ}\rangle + \langle H_{\bullet\bullet\circ}\rangle)\Big)\langle H_\bullet O\rangle. \tag{51}$$

Using (41) and (46) we then obtain

$$\langle H^4 O\rangle \sim \langle H_\bullet^4 O\rangle + 4\langle H_\circ\rangle\langle H_\bullet^3 O\rangle + 6\Big(\langle H^2\rangle - \langle H_\bullet^2\rangle - 2\langle H_\bullet\rangle\langle H_\circ\rangle\Big)\langle H_\bullet^2 O\rangle +$$

$$+4\Big(\langle H^3\rangle - \langle H_\bullet^3\rangle - 3\langle H_\circ\rangle\langle H_\bullet^2\rangle - 3\langle H_\bullet\rangle\langle H^2\rangle + 3\langle H_\bullet\rangle\langle H_\bullet^2\rangle + 6\langle H_\bullet\rangle^2\langle H_\circ\rangle\Big)\langle H_\bullet O\rangle. \tag{52}$$

We are now in a position to check the fourth local identity, which turns out to be satisfied

$$\langle H^{(4)}O\rangle = \langle H^4 O\rangle - 4\langle H\rangle\langle H^3 O\rangle - 6(\langle H^2\rangle - 2\langle H\rangle^2)\langle H^2 O\rangle - 4(\langle H^3\rangle - 6\langle H^2\rangle\langle H\rangle +$$

$$+6\langle H\rangle^3)\langle HO\rangle \sim \langle H_\bullet^4 O\rangle - 4\langle H_\bullet\rangle\langle H_\bullet^3 O\rangle - 6(\langle H_\bullet^2\rangle - 2\langle H_\bullet\rangle^2)\langle H_\bullet^2 O\rangle - 4(\langle H_\bullet^3\rangle - 6\langle H_\bullet^2\rangle\langle H_\bullet\rangle +$$

$$6\langle H_\bullet\rangle^3)\langle H_\bullet O\rangle = \langle H_\bullet^{(4)}O\rangle. \tag{53}$$

## A.1 Generic case

In order to ease the notations, we define

$$H_j = \begin{cases} (H_{j-1})_\bullet & j > 0 \\ H & j = 0, \end{cases} \tag{54}$$

which also implies $(H_k)_\circ = H_{k-1} - H_k$.

We claim

$$\langle H^n O\rangle \sim \langle H_1^n O\rangle +$$

$$+ \sum_{\substack{\{j\}_k \\ j_m > 0 \\ \sum_m j_m < n-1}} \frac{\binom{n}{\sum_m j_m}}{j_1! j_2! \cdots j_k!} \langle\{\underbrace{H - H_1}_{j_1},\underbrace{H_1 - H_2}_{j_2},\ldots,\underbrace{H_{k-1} - H_k}_{j_k}\}\rangle \langle H_{k+1}^{n-\sum_m j_m} O\rangle +$$

$$+ \sum_{\substack{\{j\}_k \\ j_m > 0 \\ \sum_m j_m = n-1}} \frac{n}{j_1! j_2! \cdots j_k!} \langle\{\underbrace{H - H_1}_{j_1},\underbrace{H_1 - H_2}_{j_2},\ldots,\underbrace{H_{k-1} - H_k}_{j_k}\}\rangle \langle H_1 O\rangle. \tag{55}$$

One can convince oneself of the validity of this equation by tracking how a generic term of the expansion is generated. As explicitly done for $n = 1, 2, 3, 4$, if we aim at factorising $\langle H_k^m O\rangle$ out of $\langle H^n O\rangle$, we must consider terms of the expansion $\langle(H_\bullet + H_\circ)^n O\rangle$ with at least $m$ operators $H_\bullet$. The procedure is then to expand again and again terms with multiple bullets (*e.g.*

$H_\bullet = H_{\bullet\circ} + H_{\bullet\bullet}$), dropping all the terms where the support of all the operators does not include the support of $O$, until the expression can be factorised in such a way that only $H_k^m$ remains attached to $O$. This is possible only if the remaining $n - m$ terms are of the form $H_i - H_{i+1}$, with $i$ running from 0 to $k - 2$. In addition, no term can be missing or the correlator would have been already disconnected, in contrast to the fact that the first factorising term can not have other than $\bullet$s. The various terms in (55) can be generated by expanding $\langle [H_k + (H_{k-1} - H_k) + \ldots + (H - H_1)]^n O \rangle$ through the formula

$$\left( \sum_{i=1}^{k} A_i \right)^n = \sum_{\substack{\{j\} \\ \sum_m j_m = n}} \frac{1}{j_1! j_2! \cdots} \{ \underbrace{A_1}_{j_1}, \underbrace{A_2}_{j_2}, \ldots \}, \tag{56}$$

which provides the nonzero coefficients of (55). Finally, the last term in (55) has been isolated to exploit the first local identity $\langle H_{k+1} O \rangle = \langle H_1 O \rangle$.

The correlators between powers of $H_{k+1}$ with $k > 0$ and $O$ on the right hand side of (55) can be worked out by replacing $H$ with $H_i$ and inverting (55) as follows

$$\langle H_{i+1}^n O \rangle \sim \langle H_i^n O \rangle -$$

$$- \sum_{\substack{\{j\}_k \\ j_m > 0 \\ \sum_m j_m < n-1}} \frac{\binom{n}{\sum_m j_m}}{j_1! j_2! \cdots j_k!} \langle \{ \underbrace{H_i - H_{i+1}}_{j_1}, \underbrace{H_{i+1} - H_{i+2}}_{j_2}, \ldots, \underbrace{H_{i+k-1} - H_{i+k}}_{j_k} \} \rangle \langle H_{i+k+1}^{n - \sum_m j_m} O \rangle -$$

$$- \sum_{\substack{\{j\}_k \\ j_m > 0 \\ \sum_m j_m = n-1}} \frac{n}{j_1! j_2! \cdots j_k!} \langle \{ \underbrace{H_i - H_{i+1}}_{j_1}, \underbrace{H_{i+1} - H_{i+2}}_{j_2}, \ldots, \underbrace{H_{i+k-1} - H_{i+k}}_{j_k} \} \rangle \langle H_1 O \rangle . \tag{57}$$

Equations (55) and (57) allow one to express $\langle H^n O \rangle$ as a linear combination of $\langle H_1^m O \rangle$, with $m \leq n$.

The next step is to recover the operators $H^{(n)}$. This can be done using (27). To that aim, it is convenient to formally rewrite (57) in exponential form

$$\langle e^{\beta H_i} O \rangle - \langle e^{\beta H_{i+1}} O \rangle \sim$$

$$\sim \sum_{n=1}^{\infty} \beta^n \sum_{\substack{\{j\}_k \\ j_m > 0 \\ \sum_m j_m < n}} \frac{\langle \{ \underbrace{H_i - H_{i+1}}_{j_1}, \underbrace{H_{i+1} - H_{i+2}}_{j_2}, \ldots, \underbrace{H_{i+k-1} - H_{i+k}}_{j_k} \} \rangle}{j_1! j_2! \cdots j_k! (n - \sum_m j_m)! (\sum_m j_m)!} \langle H_{i+k+1}^{n - \sum_m j_m} O \rangle =$$

$$= \sum_{k=1}^{\infty} \sum_{\substack{\{j\}_k \\ j_m > 0}} \beta^{\sum_m j_m} \frac{\langle \{ \underbrace{H_i - H_{i+1}}_{j_1}, \underbrace{H_{i+1} - H_{i+2}}_{j_2}, \ldots, \underbrace{H_{i+k-1} - H_{i+k}}_{j_k} \} \rangle}{j_1! j_2! \cdots j_k! (\sum_m j_m)!} \langle e^{\beta H_{i+k+1}} O \rangle . \tag{58}$$

We note that this expression is only formally correct, indeed the subsystems that allow for the canonical decomposition depend on the specific $n$ in (55) (in general, the larger $n$ and the larger the subsystems are). We can resolve this subtlety by replacing $H_m$, with $m \geq N$, by $H_{N-1}$. This choice regularises (58) without affecting $H^{(m)}$ with $m \leq N$.

Assuming this regularisation, (58) can be read as an eigenvalue equation: the truncated vector with coordinates $\langle e^{\beta H_{n-1}} O \rangle$ $(n = 1, \ldots, N)$ is an eigenvector with eigenvalue 1 of the

$N$-by-$N$ matrix

$$M_{\ell n}^{(N)}(\beta) = \begin{cases} \delta_{n2} + \sum_{\substack{\{j\}_{n-2} \\ j_m > 0}} \beta^{\sum_m j_m} \dfrac{\langle\{\underbrace{H-H_1}_{j_1},\underbrace{H_1-H_2}_{j_2},...,\underbrace{H_{n-3}-H_{n-2}}_{j_{n-2}}\}\rangle}{j_1! j_2! \cdots j_{n-2}! (\sum_m j_m)!} & \ell = 1 \\[2em] \delta_{\ell,n+1} - \sum_{\substack{\{j\}_{n-\ell} \\ j_m > 0}} \beta^{\sum_m j_m} \dfrac{\langle\{\underbrace{H_{\ell-2}-H_{\ell-1}}_{j_1},\underbrace{H_{\ell-1}-H_\ell}_{j_2},...,\underbrace{H_{n-3}-H_{n-2}}_{j_{n-\ell}}\}\rangle}{j_1! j_2! \cdots j_{n-\ell}! (\sum_m j_m)!} & \ell > 1. \end{cases} \tag{59}$$

The rest of the section is organised in a lemma-proof structure that will allow us to complete the proof of Lemma 1.

**Lemma 2.** *In a state with a finite correlations length the following equivalence is satisfied:*

$$\langle H_i^n \rangle \approx \langle H_{i+1}^n \rangle + \sum_{\substack{\{j\}_k \\ j_m > 0 \\ \sum_m j_m \le n}} \frac{\binom{n}{\sum_m j_m}}{j_1! j_2! \cdots j_k!} \langle\{\underbrace{H_i - H_{i+1}}_{j_1},\ldots,\underbrace{H_{i+k-1}-H_{i+k}}_{j_k}\}\rangle \langle H_{i+k+1}^{n-\sum_m j_m}\rangle. \tag{60}$$

*Proof of Lemma 1.* We note that the eigenspace corresponding to the eigenvalue 1 of the matrix $M^{(N)}(\beta)$ in (59) is generically nondegenerate, indeed (55) allows one to express $\langle H^n O \rangle$ in terms of $\langle H_1^m O \rangle$ without ambiguities. In exponential form (*cf.* (27)), Lemma 1 states

$$\langle e^{\beta H_n} O \rangle \sim \frac{\langle e^{\beta H_1} O \rangle}{\langle e^{\beta H_1} \rangle} \langle e^{\beta H_n} \rangle. \tag{61}$$

Thus, Lemma 1 implies that the vector with coordinates $\langle e^{\beta H_n} \rangle$ is an eigenvector of $M^{(N)}(\beta)$ with eigenvalue 1. Being the corresponding eigenspace nondegenerate, the implication holds true also in the opposite direction. By exponentiating (60) we find

$$\langle e^{\beta H_i} \rangle \sim \langle e^{\beta H_{i+1}} \rangle + \sum_{\substack{\{j\}_k \\ j_m > 0}} \beta^{\sum_m j_m} \frac{\langle\{\underbrace{H_i - H_{i+1}}_{j_1},\ldots,\underbrace{H_{i+k-1}-H_{i+k}}_{j_k}\}\rangle}{j_1! j_2! \cdots j_k! (\sum_m j_m)!} \langle e^{\beta H_{i+k+1}} \rangle, \tag{62}$$

therefore Lemma 1 is a direct consequence of Lemma 2.

$\square$

**Lemma 3.** *For any set of operators $A_j$, we have*

$$A_1^n - A_2^n = \sum_{\substack{\{j\}_k \\ j_m > 0 \\ \sum_m j_m \le n}} \frac{\{\underbrace{A_1 - A_2}_{j_1},\ldots,\underbrace{A_k - A_{k+1}}_{j_k}, \underbrace{A_{k+2}}_{n-\sum_m j_m}\}}{j_1! j_2! \cdots j_k! (n - \sum_m j_m)!}. \tag{63}$$

*Proof of Lemma 2.* Since the correlation length is finite, we can merge back the expectation values in (60)

$$\langle H_i^n \rangle \sim \langle H_{i+1}^n \rangle + \sum_{\substack{\{j\}_k \\ j_m > 0 \\ \sum_m j_m \le n}} \frac{\langle\{\underbrace{H_i - H_{i+1}}_{j_1},\ldots,\underbrace{H_{i+k-1}-H_{i+k}}_{j_k},\underbrace{H_{i+k+1}}_{n-\sum_m j_m}\}\rangle}{j_1! j_2! \cdots j_k! (n - \sum_m j_m)!}. \tag{64}$$

By Lemma 3, this is in fact an identity, valid independently of the operators $H_i$.

$\square$

*Proof of Lemma 3.* When we exponentiate (63) we end up with

$$e^{xA_1} - e^{xA_2} = \sum_{k=1}^{\infty} \sum_{\substack{\{j\}_k \\ j_m > 0}} \sum_{n=0}^{\infty} \frac{\overbrace{\{xA_1 - xA_2, \ldots, \underbrace{xA_k - xA_{k+1}}_{j_k}, \underbrace{xA_{k+2}}_{n}\}}^{j_1}}{n! j_1! j_2! \cdots j_k! (n + \sum_m j_m)!}, \tag{65}$$

where the regularisation $A_m = A_{N-1}$ for $m \geq N$ is understood. Led by the symmetry of the products, we conjecture that (65) can be written as[9]

$$e^{xA_1} - e^{xA_2} = \sum_{k=1}^{N-2} \sum_{\substack{\{s\}_k \\ s_m = \pm 1}} \Big(\prod_{j=1}^{k} s_j\Big) e^{xA_{k+2} + \sum_{j=1}^{k} \frac{1+s_j}{2} x(A_j - A_{j+1})}, \tag{66}$$

where the sum over $k$ has been truncated to $N-2$ by virtue of the regularisation. We readily see that the contribution from a sequence $\{s_1, \ldots, s_k\}$ is cancelled out by the one from the sequence of length $N-2$ with elements $\{s_1, \ldots, s_k, -1, 1, \ldots, 1\}$. Thus, only the terms with $s_m = 1 \, \forall m > 1$ remain. They correspond to the left hand side of (66), proving in turn its validity. If the conjecture on the basis of (66) is correct, we can conclude that (63) holds true independently of whether the operators $A_j$ commute or not. In order to dispel any doubt, we have also confirmed (63) up to $n = 6$ for generic non-commuting operators using Mathematica. $\qquad\square$

Although we considered spin chains, the proof of Lemma 1 seems to be easily generalisable to higher dimensions, so the lemma is expected to hold also for $d > 1$.

In conclusion, as far as quasilocal Hamiltonians are considered, the main assumption of this paper can be broken only when the initial state has power-law decaying correlations.

## B  Alternative averages

In this appendix we consider nonuniform time averages

$$\bar{\boldsymbol{\rho}}_{t_0, t} = \int_{t_0}^{t_0 + t} \mathrm{d}\tau \, \wp_t(\tau - t_0) |\Psi_\tau\rangle \langle \Psi_\tau|, \tag{67}$$

where $\wp_t$ is a probability distribution in $[0, t]$. The asymptotic behaviour of the moments of $\bar{\boldsymbol{\rho}}_{t_0, t}$ can be carried out as in the uniform case. Specifically, we have

$$\mathrm{tr}[\bar{\boldsymbol{\rho}}_t^\alpha] \sim \int \cdots \int_{[0, t L^{\frac{d}{2}}]^\alpha} \frac{\mathrm{d}^\alpha \tau}{L^{d\frac{\alpha}{2}}} \Big(\prod_{j=1}^{\alpha} \wp_t(L^{-\frac{d}{2}} \tau_j)\Big) e^{-\mathfrak{e}_2 \frac{(\tau_\alpha - \tau_1)^2 + \sum_{j=1}^{\alpha-1}(\tau_j - \tau_{j+1})^2}{2}}. \tag{68}$$

Let us change variables into

$$\tau_j' = \begin{cases} \tau_j - \tau_{j+1} & j < \alpha \\ L^{-\frac{d}{2}} \tau_\alpha & j = \alpha, \end{cases} \tag{69}$$

where we rescaled (back) $\tau_\alpha$ because it does not appear in the gaussian anymore; we find

$$\mathrm{tr}[\bar{\boldsymbol{\rho}}_t^\alpha] \sim \frac{1}{L^{d\frac{\alpha-1}{2}}} \int_0^t \mathrm{d}\tau_\alpha' \int_{-\tau_\alpha' L^{\frac{d}{2}}}^{(t - \tau_\alpha') L^{\frac{d}{2}}} \mathrm{d}\tau_{\alpha-1}' \int_{-\tau_\alpha' L^{\frac{d}{2}} - \tau_{\alpha-1}'}^{(t - \tau_\alpha') L^{\frac{d}{2}} - \tau_{\alpha-1}'} \mathrm{d}\tau_{\alpha-2}' \cdots \int_{-\tau_\alpha' L^{\frac{d}{2}} - \sum_{j=2}^{\alpha-1} \tau_j}^{(t - \tau_\alpha') L^{\frac{d}{2}} - \sum_{j=2}^{\alpha-1} \tau_j} \mathrm{d}\tau_1'$$

$$\Big(\prod_{j=1}^{\alpha} \wp_t(\tau_\alpha' + L^{-\frac{d}{2}} \sum_{n=j}^{\alpha-1} \tau_n')\Big) e^{-\mathfrak{e}_2 \frac{(\sum_{j=1}^{\alpha-1} \tau_j')^2 + \sum_{j=1}^{\alpha-1}(\tau_j')^2}{2}}. \tag{70}$$

---

[9]This is clearly true if the matrices commute with one another.

Since the gaussian forces all the variables $\tau'_j$ with $j \in 1, \ldots, \alpha - 1$ to be $O(1)$, we can extend their integration domain to infinity; in addition, at the leading order, they also disappear from the argument of $\wp_t$

$$\text{tr}[\bar{\boldsymbol{\rho}}_t^\alpha] \sim \int_0^t d\tau'_\alpha [\wp_t(\tau'_\alpha)]^\alpha \int \cdots \int_{[-\infty,\infty]^{\alpha-1}} \frac{d^{\alpha-1}\tau'}{L^{d\frac{\alpha-1}{2}}} e^{-\mathfrak{e}_2 \frac{(\sum_{j=1}^{\alpha-1} \tau'_j)^2 + \sum_{j=1}^{\alpha-1}(\tau'_j)^2}{2}} =$$

$$= \alpha^{-\frac{1}{2}} \left(\frac{\mathfrak{e}_2}{2\pi}\right)^{\frac{1-\alpha}{2}} L^{d\frac{1-\alpha}{2}} \int_0^t d\tau [\wp_t(\tau)]^\alpha. \quad (71)$$

From this, we readily obtain the Rényi entropies

$$S_\alpha[\bar{\boldsymbol{\rho}}_t] = \frac{d}{2} \log L + \frac{1}{2} \log \frac{\mathfrak{e}_2 t^2}{2\pi} + \frac{\log \alpha}{2(\alpha-1)} + \frac{1}{1-\alpha} \log \int_0^t d\tau [\wp_t(\tau)]^\alpha + O(L^{-\frac{d}{2}}) \quad (72)$$

and the von Neumann entropy

$$S_{\text{vN}}[\bar{\boldsymbol{\rho}}_t] \sim \frac{d}{2} \log L + \frac{1}{2} \log \frac{\mathfrak{e}_2}{2\pi} + \frac{1}{2} - \int_0^t d\tau \wp_t(\tau) \log \wp_t(\tau). \quad (73)$$

We note that the von Neumann entropy is maximal when $\wp_t(\tau)$ is uniform ($\wp_t(\tau) = \frac{1}{t}$), suggesting (as expected) that the effective dimension $\mathfrak{D}_t^{(\epsilon_t)}$ is maximised by the uniform average.

The distribution of eigenvalues can be computed with the method reported in the main text, and we find

$$\Phi_{\bar{\boldsymbol{\rho}}_t}(\lambda) \sim \int_0^t d\tau \frac{L^{\frac{d}{2}}}{\pi} \sqrt{\frac{\mathfrak{e}_2}{\log \frac{2\pi[\wp_t(\tau)]^2}{\mathfrak{e}_2 L^d \lambda^2}}} \theta_H \left(\wp_t(\tau) - \sqrt{\frac{\mathfrak{e}_2}{2\pi}} \lambda L^{\frac{d}{2}}\right). \quad (74)$$

We point out that the change of scale in the eigenvalues

$$p = \Omega \lambda, \qquad \text{with} \quad \Omega = \sqrt{\frac{\mathfrak{e}_2}{2\pi}} L^{\frac{d}{2}}, \quad (75)$$

brings the distribution into a universal form

$$d\lambda \Phi_{\bar{\boldsymbol{\rho}}_t}(\lambda) \sim dp \int_0^t \frac{d\tau}{\sqrt{\pi \log \frac{\wp_t(\tau)}{p}}} \theta_H \left(\wp_t(\tau) - p\right). \quad (76)$$

Finally, the dimension $\mathfrak{D}_t^{(\epsilon_t)}$ of the "weighted" space visited by the state is the solution to the following system

$$\epsilon_t = \Omega \int_0^t d\tau \wp_t(\tau) \left[1 - \text{erf}\left(\sqrt{-\log \min\left(\frac{p_{\epsilon_t}}{\wp_t(\tau)}, 1\right)}\right)\right]$$

$$\mathfrak{D}_t^{(\epsilon_t)} = \frac{2\Omega^2}{\sqrt{\pi}} \int_0^t d\tau \theta_H (\min(\wp_t(\tau), \Omega) - p_{\epsilon_t}) \left[\sqrt{\log \frac{\wp_t(\tau)}{p_{\epsilon_t}}} - \sqrt{\log \min\left(\frac{\wp_t(\tau)}{\Omega}, 1\right)}\right]. \quad (77)$$

## C  Leading correction

In this appendix we work out the leading correction to the asymptotic behaviour of the moments $\text{tr}[\bar{\boldsymbol{\rho}}_t^\alpha]$ when the number of the sites is large. As suggested by (8), that is a relative

correction $O(L^{-\frac{d}{2}})$ that comes from the constraint on the integration variables, which have to sum to zero. From the representation (11) it follows that the higher order cumulants start contributing at relative order $O(L^{-d})$, therefore our starting point is the second line of (12). The leading correction becomes visible if we integrate first in $\tau'_\alpha$. To that aim, we must move the first integral to the right, like we did in (8). The result is

$$\int\cdots\int_{\tau'_{\alpha-j}\in[-tL^{\frac{d}{2}}+\max(T_j),tL^{\frac{d}{2}}+\min(T_j)]}\frac{d^{\alpha-1}\tau'}{t^{\alpha-1}L^{d\frac{\alpha-1}{2}}}\Big(1+\frac{\min(T_\alpha)-\max(T_\alpha)}{tL^{\frac{d}{2}}}\Big)e^{-\mathfrak{e}_2\frac{(\sum_{j=1}^{\alpha-1}\tau'_j)^2+\sum_{j=1}^{\alpha-1}(\tau'_j)^2}{2}},\quad(78)$$

where $T_1=\{0\}$ and $T_j=T_{j-1}\cup\{-\sum_{n=1}^{j-1}\tau'_{\alpha-n}\}$. The leading term, which we already computed, comes from the 1 in the round bracket; the other term includes the leading correction. The domain of integration is rather complicated, but, for an asymptotically large number of sites, it can be extended to infinity. In addition, by reverting the sign of all $\tau'_j$, we realise that the contribution from the term proportional to $\min(T_\alpha)$ is equal to the contribution from the one proportional to $-\max(T_\alpha)$. Thus we have

$$\mathrm{tr}[\bar{\boldsymbol{\rho}}_t^\alpha]-\alpha^{-\frac{1}{2}}\Big(\frac{\mathfrak{e}_2}{2\pi}\Big)^{\frac{1-\alpha}{2}}t^{1-\alpha}L^{d\frac{1-\alpha}{2}}\sim-2\int\cdots\int_{[-\infty,\infty]^{\alpha-1}}\frac{d^{\alpha-1}\tau}{t^\alpha L^{d\frac{\alpha}{2}}}\max(T_\alpha)e^{-\mathfrak{e}_2\frac{(\sum_{j=1}^{\alpha-1}\tau'_j)^2+\sum_{j=1}^{\alpha-1}\tau'^2_j}{2}}.\quad(79)$$

This expression can be simplified further by defining the new variables

$$y_j=\sqrt{\mathfrak{e}_2}\sum_{n=j}^{\alpha-1}\tau'_j\quad(80)$$

and summing over the $\alpha-1$ possibilities for which $\max(T_\alpha)=\mathfrak{e}_2^{-1/2}\max(0,y_j)$. We finally obtain

$$\mathrm{tr}[\bar{\boldsymbol{\rho}}_t^\alpha]-\alpha^{-\frac{1}{2}}\Big(\frac{\mathfrak{e}_2}{2\pi}\Big)^{\frac{1-\alpha}{2}}t^{1-\alpha}L^{d\frac{1-\alpha}{2}}\sim$$
$$\sim-2\mathfrak{e}_2^{-\frac{\alpha}{2}}t^{-\alpha}L^{-d\frac{\alpha}{2}}\int\cdots\int_{[0,\infty]^{\alpha-1}}d^{\alpha-1}y\sum_{j=1}^{\alpha-1}y_1e^{-\frac{y_1^2+\sum_{j=1}^{\alpha-2}(y_j-y_{j+1})^2+y_{\alpha-1}^2}{2}}.\quad(81)$$

We have not found a closed form expression for the gaussian integral on the right hand side of the equation (already for $\alpha=4$ we end up with an integral of the error function). This makes it trickier to compute the leading correction in the distribution of eigenvalues, which we leave to future investigations.

# D  Numerical checks

In this appending we report some numerical checks of our findings in spin chains ($d=1$).

**Transverse field Ising chain.**  The transverse field Ising chain is described by the Hamiltonian

$$H(h)=-J\sum_\ell\Big(\boldsymbol{\sigma}_\ell^x\boldsymbol{\sigma}_{\ell+1}^x+h\boldsymbol{\sigma}_\ell^z\Big),\quad(82)$$

where $\boldsymbol{\sigma}_\ell^\alpha\equiv\cdots\otimes I_{\ell-2}\otimes I_{\ell-1}\otimes\sigma_\ell^\alpha\otimes I_{\ell+1}\otimes I_{\ell+2}\otimes\cdots$ acts like the Pauli matrix $\sigma_\ell^\alpha$ ($\alpha\in\{x,y,z\}$) on site $\ell$ and like the identity elsewhere. A Jordan-Wigner transformation maps the spin chain

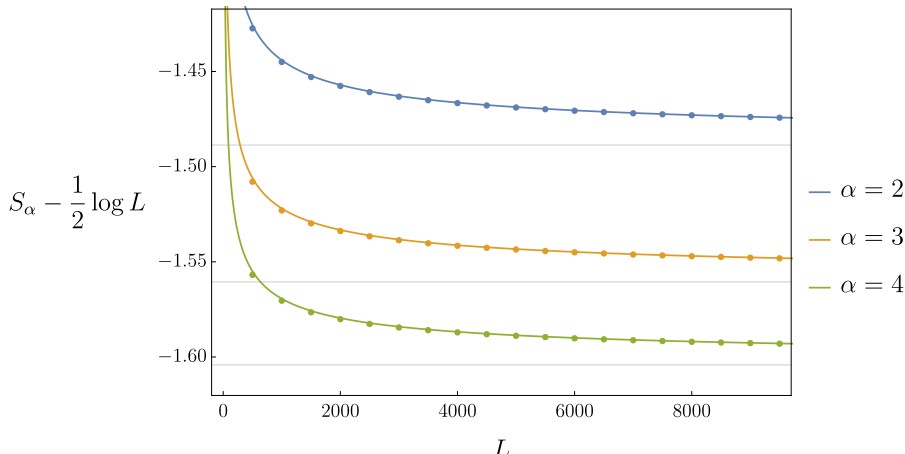

Figure 3: The Rényi entropies $S_\alpha$, with $\alpha = 2, 3, 4$, of the time averaged state in the time window $[0, 0.4J^{-1}]$ after a quench of the magnetic field $h = \infty \to 1.5$ in the transverse field Ising chain (82). The dots correspond to numerical evaluations of (11). The curves are the asymptotic predictions plus the leading correction computed in Appendix C. The horizontal lines show the limit $L \to \infty$ (which are the predictions without the leading correction).

into a chain of fermions, and the resulting Hamiltonian consists of two sectors where it acts like a quadratic form. This allows for exact diagonalization, and also for the exact solution of the dynamics when the initial state is a Slater determinant. For example, one can easily compute the function $f(t)$ defined in (7) after a global quench of $h : h_i \to h_f$ (the system is prepared in the ground state of $\boldsymbol{H}(h_i)$ and then let to evolve under $\boldsymbol{H}(h_f)$) [9]

$$f(t) = \int_0^\pi \frac{dk}{2\pi} \log\Big(\frac{1 + \cos\Delta_k}{2} + \frac{1 - \cos\Delta_k}{2} e^{2i\varepsilon_k t}\Big), \tag{83}$$

where

$$\varepsilon_k = 2J\sqrt{1 + h_f^2 - 2h_f \cos k}$$

$$\cos\Delta_k = \frac{(h_f - \cos k)(h_i - \cos k) + \sin k^2}{\sqrt{1 + h_f^2 - 2h_f \cos k}\sqrt{1 + h_i^2 - 2h_i \cos k}}. \tag{84}$$

We are therefore in a position to check our predictions for the first Rényi entropies of $\bar{\boldsymbol{\rho}}_t$ in the limit of large $L$. Figure 3 shows a comparison between the numerical evaluation of the entropies[10] (as logarithms of multidimensional integrals) and our asymptotic predictions. The agreement is excellent.

**Exact diagonalization.** Despite giving access to the first Rényi entropies, the exactly solvable model considered in the previous paragraph does not allow us to easily check the dimension of the relevant subspace. To overcome this problem, we have carried out a numerical analysis based on exact diagonalization algorithms in small spin chains with rather generic Hamiltonians. In principle our prediction is not expected to be accurate, as it was derived in the opposite limit of large $L$; the agreement between numerical data and prediction is nevertheless surprisingly good, as shown in figures 4 and 6.

---

[10]As a matter of fact, we have dropped some exponentially small finite-size effects by replacing a sum by an integral in the logarithm of the overlap, *i.e.*, by writing $\langle \Psi_0 | e^{iHt} | \Psi_0 \rangle = \exp(Lf(t))$, with $f(t)$ given by (83).

We also checked that our Ansatz $\epsilon_t \sim \epsilon_{\delta t}\sqrt{\delta t/t}$ generates a subspace approximating the time evolving state with an accuracy that increases with the time. To that aim, we have computed

$$\text{error}_{(L)}(t, T) = 1 - \langle \Psi_t | \theta_H(\bar{\boldsymbol{\rho}}_T - \lambda_{\epsilon_T}) | \Psi_t \rangle \qquad t \in [0, T], \tag{85}$$

which is the error made by projecting the state at the time $t$ onto the subspace corresponding to $\epsilon_T$. As shown in figures 5 and 7, the numerical data confirm that this Ansatz provides an upper bound to the dimension of the relevant space. In all the examples that we considered, the maximal error is associated with the boundaries of the interval.

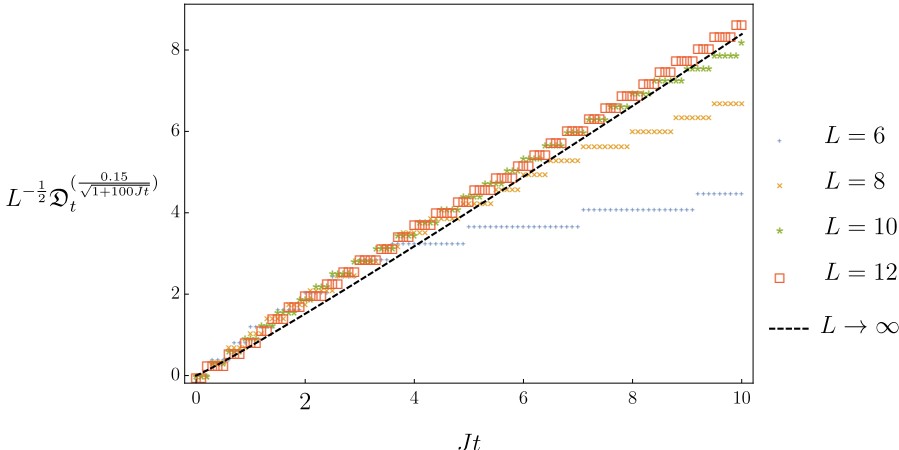

Figure 4: The minimal number of eigenstates per unit $\sqrt{L}$ with probability larger or equal to $1 - \epsilon_t$, with $\epsilon_t = {}^{0.15}/\sqrt{1+100Jt}$ in a small spin-$\frac{1}{2}$ chain with $L = 6, 8, 10, 12$ sites, obtained using exact diagonalization techniques. The initial state is the ground state of the ferromagnetic Ising Hamiltonian $\boldsymbol{H}_0 = -J \sum_\ell (\sigma^x_\ell \sigma^x_{\ell+1} + 2\sigma^y_\ell)$; time evolution is generated by $\boldsymbol{H} = J \sum_\ell \sigma^y_\ell \sigma^y_{\ell+1} + 0.5\sigma^x_\ell \sigma^x_{\ell+1} + 1.5\sigma^z_\ell \sigma^z_{\ell+1} + 0.25\sigma^x_\ell + 0.3(-1)^\ell \sigma^z_\ell$. The dashed line is the prediction (19). Data seem to collapse to the prediction rather quickly.

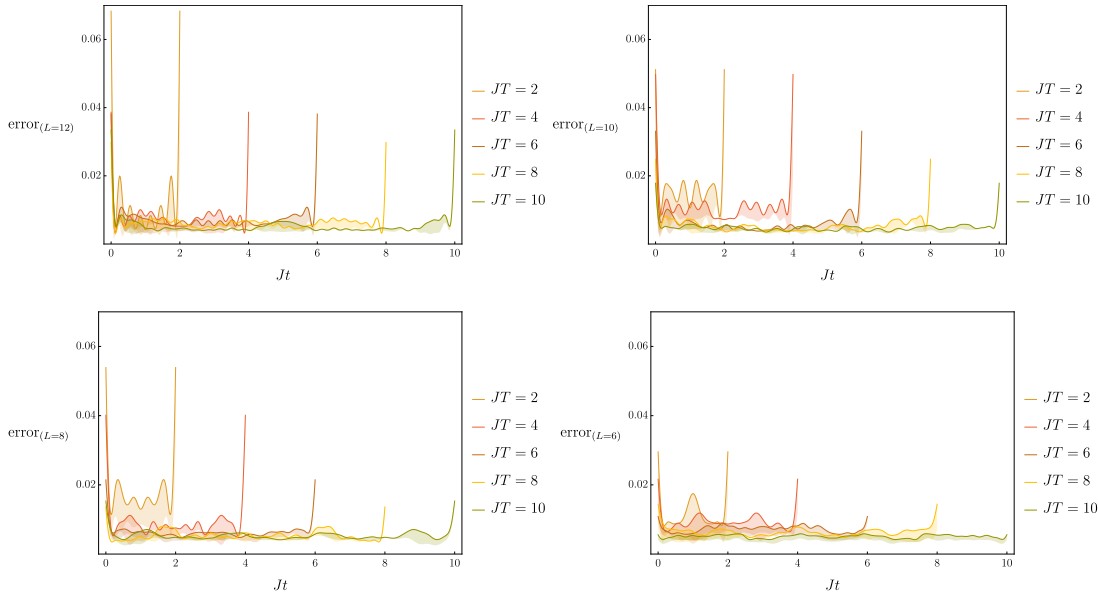

Figure 5: The error on the state at time $t$ induced by the Ansatz $\epsilon_t \sim t^{-\frac{1}{2}}$ (specifically, $\epsilon_t = {}^{0.15}/\sqrt{1+100Jt}$) for the same parameters as in figure 4 in chains with $L = 12, 10, 8, 6$ sites. Each curve corresponds to projecting onto the reduced space associated with the time window $[0, T]$. The shades below the curves represent an indetermination coming from the fact that $\epsilon_t$ can be enforced only approximately because the error made by reducing the space is in fact quantised.

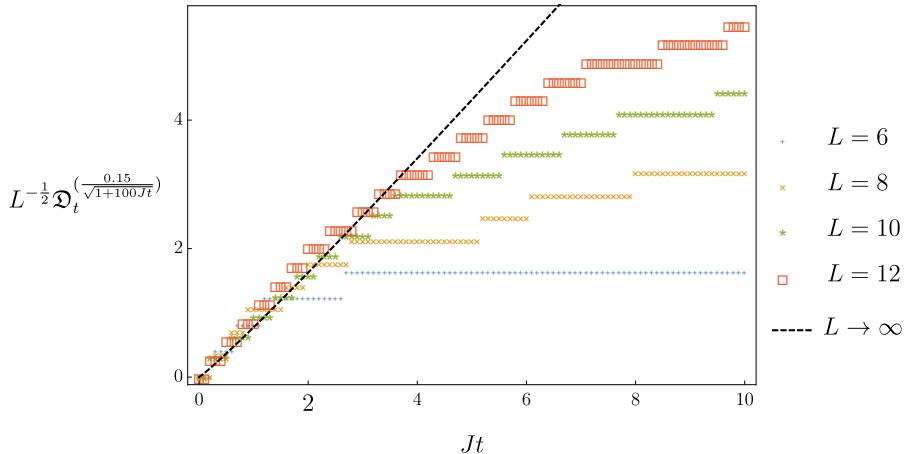

Figure 6: The same as in figure 4 but for a different system. Here the initial state is fully polarised in the $z$ direction and time evolution is generated by $H = J \sum_\ell \sigma_\ell^x \sigma_{\ell+1}^x + 2\sigma_\ell^y \sigma_{\ell+1}^y + \sigma_\ell^z \sigma_{\ell+1}^z$, which describes an integrable system.

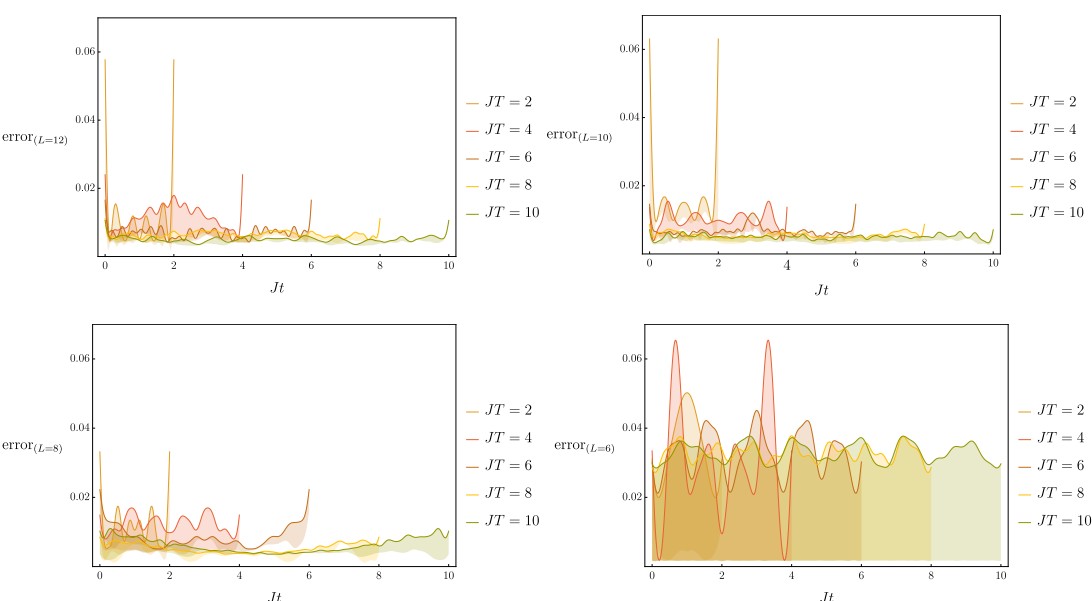

Figure 7: The same as in figure 5 for the system of figure 6.

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
