# Peer review of "On the size of the space spanned by a nonequilibrium state in a quantum spin lattice system"

_SciPost Physics, doi:SciPost Phys. 6, 059 (2019)_

## Round 2 · Referee Report · Anonymous · 2019-4-8

Strengths

1. analytical results for prototypical problem (quantum quench), derived in situation where thermodynamic limit is taken first
2. explicit check for specific cases (e.g., quantum Ising chain)

Weaknesses

1. assumptions underlying the derivation unclear
2. practical relevance not very clear

Report

It is known that if a quantum-mechanical system of a fixed, finite size L is prepared in a pure state and then time-evolved with a given Hamiltonian, the long-time limit of the time-averaged density matrix is described by the so-called diagonal ensemble. In his paper "On the size of the space spanned by a nonequilibrium state in a quantum spin lattice system", the author studies the same setup but focuses on the opposite limit: The author investigates the behavior of the density matrix averaged over a given, fixed time window in the limit of infinite system size. This is an interesting question in its own right which has not received much attention.

First, very general (model-independent) arguments are given based on series expansions evaluated in the limit of large L. Analytical expressions for the distribution of eigenvalues of the time-averaged density matrix and from that the "rank" of the time-averaged density matrix are computed. These results are later (in the appendix) compared with explicit data obtained analytically for the 1d quantum Ising model or numerically using exact diagonalization.

I do think that these results can be published. However, I do have several suggestions (see below).

Requested changes

1) I did not quite understand under which assumptions the results presented in the main text are valid. E.g., are there any assumptions about the nature of the spectrum of the Hamiltonian that governs the time evolution (gapless vs. gapped)?

2) Directly related to 1): Would the explicit example of the quantum Ising chain (Figure 3) also work if the quench ended at the critical point h=1 or in the other phase h<1?

3) I would suggest to not hide the comparison with the explicit calculations in the appendix but to discuss them in the main text.

4) I did not quite understand what the practical relevance of the results is - some more comments would be helpful. In particular, can these results be used to infer something about the behaviour of local observables?

  • validity: high
  • significance: good
  • originality: high
  • clarity: good
  • formatting: excellent
  • grammar: excellent

Author:  Maurizio Fagotti  on 2019-04-17  [id 498]

(in reply to Report 1 on 2019-04-08)
Category:
remark
answer to question

I thank the referee for reading the paper and for making useful suggestions. Before commenting on the changes, I’d like to dispel the weaknesses pointed out by the referee.

The first weakness was about the assumptions, which the referee found unclear. Essentially, this paper is based on a single assumption, i.e., the energy cumulants being proportional to the number of the lattice sites $L^d$. No additional hypothesis seems to be required in order to carry out the asymptotic expansion in the limit of large $L$. The referee seems to be concerned about which physical systems have this property, possibly suspecting them to be somehow exceptional. As a matter of fact, the basic assumption has the same degree of generality of saying that the free energy of a thermal system is proportional to the volume and is a smooth function of the inverse temperature. Exceptions to this rule are very well known and gave rise to the theory of phase transitions and critical phenomena; nevertheless, they are still exceptions. Analogously, it is possible to find exceptional systems that do not meet the hypothesis; being this the first investigation on the subject, I opted for presenting the generic situation, leaving the analysis of exceptions to future works. That said, I agree with the referee that the previous version of the manuscript did not provide any evidence in support of this claim of generality. Unfortunately, I did not find a good reference generalising the work by Griffiths (Journal of Mathematical Physics 5, 1215 (1964)), focussed on classical spin systems in thermal equilibrium, to the quantum nonequilibrium setting considered in this work. Thus, I am going to include an appendix with a proof that the cumulants are extensive, provided that the Hamiltonian is (quasi)local and the initial state has a finite correlation length. This answers some of the referee’s questions; in particular, it does not matter whether the Hamiltonian is critical or not, but it does matter whether the initial state is the ground state of a critical or of a non-critical system.

The second weakness spotted by the referee is about the practical relevance of this work. I tried to go in this direction in the subsection on the numerical simulations, and the new version of the manuscript will report a practical application. Specifically, the main result can be used to get a physical reference value for the time step to choose in numerical simulations of time evolution. This turns out to be time dependent and to approach zero as $t^{-d/2}$. From a more theoretical point of view, I think that this work will become even more relevant when systems not satisfying the main assumption will be investigated: one will be then in a position to find new quantitative and qualitative differences between the nonequilibrium time evolution of critical and of non-critical systems.

In the following, I comment on the requested changes: 1) I already answered this question; I stress here that there is no assumption on the spectrum of the Hamiltonian. On the other hand, if the initial state is the ground state of a critical Hamiltonian, there could be problems. 2) Yes, it would work. 3) I prefer to keep the numerical analysis in an appendix for three reasons: - It does not add anything to the result, which is already proved analytically. - It is specific to spin chains, whereas the result holds true also in higher dimensions. - I used very basic numerical algorithms, so the numerical data could be readily obtained again (and also improved) by any interested reader. Nevertheless, I agree that that appendix was hidden, indeed there was no reference to it in the main text. The new version of the manuscript will resolve this problem. 4) I already answered this question.

---

## Round 4 · Author Response

In this version of the manuscript (v4), the upper bound on the size of the space, presented in the original submitted version (v2), is shown to be saturated. It is also proved that the assumption that the energy cumulants are extensive is almost always fulfilled, with important exceptions when the initial state has power-law decaying correlations.

---

## Round 4 · List of Changes

- An appendix (Appendix A) has been added with a proof that the cumulants of a quasilocal Hamiltonian are extensive, provided that the state has finite correlation lengths.
- Section 4 has been improved.
- Section 4.1 now includes a practical application of the main result: it provides a physical criterion to fix the time step of the numerical simulations of the dynamics.
- References to the appendices have been added in the main text.
- Some typos have been fixed.

You are currently on this page

Resubmission 1901.10797v4 on 30 April 2019

---

## Editorial Decision

published